# Phylogenetic and Comparative Analyses of Complete Chloroplast Genomes of Chinese *Viburnum* and *Sambucus* (Adoxaceae)

**DOI:** 10.3390/plants9091143

**Published:** 2020-09-03

**Authors:** Hang Ran, Yanyan Liu, Cui Wu, Yanan Cao

**Affiliations:** College of Plant Protection, Henan Agricultural University, Zhengzhou 450002, China; hnnydxrh@126.com (H.R.); liuyanyan@henau.edu.cn (Y.L.); wucui0605@163.com (C.W.)

**Keywords:** *Viburnum*, *Sambucus*, chloroplast genome, comparative genomics, phylogeny

## Abstract

Phylogenetic analyses of complete chloroplast genome sequences have yielded significant improvements in our understanding of relationships in the woody flowering genus *Viburnum* (Adoxaceae, Dipsacales); however, these relationships were evaluated focusing only on *Viburnum* species within Central and South America and Southeast Asia. By contrast, despite being a hotspot of *Viburnum* diversity, phylogenetic relationships of *Viburnum* species in China are less well known. Here, we characterized the complete chloroplast (cp) genomes of 21 *Viburnum* species endemic to China, as well as three *Sambucus* species. These 24 plastomes were highly conserved in genomic structure, gene order and content, also when compared with other Adoxaceae. The identified repeat sequences, simple sequence repeats (SSRs) and highly variable plastid regions will provide potentially valuable genetic resources for further population genetics and phylogeographic studies on *Viburnum* and *Sambucus*. Consistent with previous combined phylogenetic analyses of 113 *Viburnum* species, our phylogenomic analyses based on the complete cp genome sequence dataset confirmed the sister relationship between *Viburnum* and the *Sambucus*-*Adoxa*-*Tetradoxa*-*Sinadoxa* group, the monophyly of four recognized sections in *Flora of China* (i.e., *Viburnum* sect. *Tinus*, *Viburnum* sect. *Solenotinus*, *Viburnum* sect. *Viburnum* and *Viburnum* sect. *Pseudotinus*) and the nonmonophyly of *Viburnum* sect. *Odontotinus* and *Viburnum* sect. *Megalotinus*. Additionally, our study confirmed the sister relationships between the clade Valvatotinus and *Viburnum* sect. *Pseudotinus*, as well as between *Viburnum* sect. *Opulus* and the *Odontotinus*-*Megalotinus* group. Overall, our results clearly document the power of the complete cp genomes in improving phylogenetic resolution, and will contribute to a better understanding of plastome evolution in Chinese Adoxaceae.

## 1. Introduction

The eudicot family Adoxaceae (Dipsacales) sensu APG IV contains three small herbaceous genera (less than 10 species) (i.e., *Adoxa*, *Sinadoxa*, and *Tetradoxa*) and two larger genera (i.e., *Viburnum* and *Sambucus*) [1]. The woody flowering taxon *Viburnum*, with approximately 200 species of shrubs and small trees [2], is the largest genus within Adoxaceae, and is of great interest to the horticultural community, since more than 70 of these species (and a variety of artificial hybrids) have been brought into cultivation [3]. Although widely distributed in the Northern Hemisphere, *Viburnum* has major centers of species diversity in eastern Asia and Central and South America [4,5,6], with significant extensions into the montane forests of Southeast Asia [7] and South America [8]. *Sambucus* is a relatively small genus occurring mostly in the north temperate zone, comprising about 10 species of small trees, shrubs and perennial herbs [5,9,10], of which many species are cultivated ornamentally, and several produce edible fruits (https://www.britannica.com/plant/Dipsacales). In addition, several species are commonly used in folk medicine (e.g., *S. adnata*, *S. javanica* and *S. nigra*) [11]. Within Adoxaceae, analyses of complete cp genome sequences suggested that *Sambucus* and *Viburnum* were the most closely related [12]; more specifically, *Viburnum* was likely to be the sister group of *Sambucus* plus *Adoxa* and its relatives. Although both genera have important horticultural value, limited molecular markers were available for the application, breeding and conservation of these species in the context of population genetics and phylogenetic studies.

Based on various morphological characteristics (e.g., endocarp shape, inflorescence form, leaf morphology, the presence or absence of naked buds and of sterile flowers around the margins of the inflorescences), *Viburnum* has been subdivided by several researchers, most commonly into ten groups formally recognized as sections [5,13]. Over the past decade, great advances have been made in understanding *Viburnum* phylogeny [14,15,16,17,18]. The number of species sampled in phylogenetic studies has increased from 40 to 90, representing all major clades within the genus. Additionally, sampling has increased from four to ten genes, thus affording better phylogenetic resolution. These phylogenetic studies have uniformly and strongly supported earlier recognized sections and subsections, while encountering difficulties resolving the relationships with confidence based on limited parsimony informative sites, in particularly with recent divergences within groups of closely related species [19]. Nonetheless, a recent study of 22 species provided us, for the first time, with comparatively high-resolution data of nearly all of the deepest branching events within *Viburnum* in light of next-generation sequencing of whole plastid genomes [19]. This study demonstrated a reliable framework within which to assess the power of complete cp genome markers and methods to discriminate *Viburnum* species in Central and South America (16 species) and Southeast Asia (6 species). By contrast, China is considered to be one of the hotspots of *Viburnum* plant taxa diversity; a total of 8 sections and *c*. 73 species have been found in this region [2]. Nevertheless, the phylogenetic relationships of *Viburnum* species there have received much less attention.

In the present study, we reported whole-plastome sequence data for 21 species of *Viburnum*, covering all of the eight currently diagnosed sections in *Flora of China*, as well as for three species of *Sambucus*. The main goals of this study were to: (1) characterize and compare the cp genomes of *Viburnum* species belonging to all the eight sections occurring in China and related taxa in order to gain insights into their evolutionary patterns; (2) examine the phylogenetic relationships of the main clades of Chinese Adoxaceae, with a particular focus on the generic status of *Viburnum*; and (3) screen and identify repeat sequences, simple sequence repeats (SSRs) and mutational hotspot regions for future species identification and phylogeographic studies of the two genera.

## 2. Results and Discussion

### 2.1. Chloroplast Genome Assembly and Features

With the Illumina HiSeq 2500 system (San Diego, CA, USA), we sequenced the plastomes of 21 species of *Viburnum* and 3 species of *Sambucus*. Of these samples, through de novo assembly, the maximum number of assembled contigs ranged from 61,001 (*V. odoratissimum*) to 388,130 (*V. melanocarpum*), with N50 contigs varying from 285 to 399 bp. Average sequencing depth ranged from about 268× (*S. adnata*) to 517× (*V. melanocarpum*) (Appendix A). Subsequently, through reference-based assembly, a total of 165–209 contigs were successfully mapped to the reference plastomes. Among these, three to eight long contigs (>10 kb) that were found to be significantly homologous to the reference genome were combined to generate each chloroplast genome, with no gaps found. The four junctions between IRs and SSC/LSC in each species were initially determined on the basis of these long contigs, and then verified by PCR-based sequencing. The results showed that the assembly sequences were totally identical with the PCR amplified fragments, demonstrating the high quality of our assembly. Finally, we obtained 24 whole chloroplast genome sequences without gaps after de novo and reference-guided assembly, and submitted them to GenBank with accession numbers MT507585–MT507605 for *Viburnum* and MT457821–MT457823 for *Sambucus* (Appendix A).

The complete cp genomes of the 21 *Viburnum* species were determined to be 157,833–158,652 bp in size, and the three *Sambucus* species ranged from 158,102 bp (*S. nigra*) to 158,756 bp (*S. adnata*) (Table 1). Akin to most land plant species, all of these plastomes exhibited a typical quadripartite structure, including a pair of IR regions (26,272–26,564 bp) separating the LSC region (86,430–87,892 bp) and the SSC region (17,674–18,978 bp). The overall GC content in the whole genome sequences was practically identical among these plastomes (38.0–38.2%). The 21 *Viburnum* cp genomes encoded the same 130 functional genes, consisting of 85 protein-coding genes, 37 transfer RNA (tRNA) genes and 8 ribosomal RNA (rRNA) genes. The 3 *Sambucus* cp genomes encoded identical sets of 132 genes, with 84 protein-coding genes, 40 tRNA genes and 8 rRNA genes (Table 1). Notably, five genes (i.e., *trnM-CAU*, *trnT-GGU*, *trnP-GGG*, *orf188* and *lhbA*) and three genes (i.e., *psbZ*, *ndhH* and *rpl22*) were only present in *Sambucus* and *Viburnum*, respectively. For both genera, 15 genes possessed a single intron (nine protein-coding genes and six tRNA genes), while 3 (*ycf3*, *clpP* and *rps12*) contained two introns, and a total of 17 genes were duplicated in the IR regions (Table 1). In particular, the *rps12* was a transspliced gene, with the first exon located in the LSC region, and the second and third in the IR regions. We also found that the *ycf1* gene at the SSC and IRa junction was present as a pseudogene in 16 *Viburnum* species (Table 2), due to the incomplete gene duplication, as shown in previous reports [20,21]. In addition, there were some exceptions where non-ATG codons were translated as Met and identified as start codons, such as GCT for *psbL*, GTG for *rps19* and CTG for *ndhD*, which has also been observed in many other angiosperms, for instance, *Betula platyphylla* [22] and *Punica granatum* [23].

The newly obtained whole plastome sequences for 21 *Viburnum* species, plus three *Sambucus* taxa, vary only slightly in size (157,833–158,756 bp) (Table 1), and are greatly similar in overall structure, gene content and arrangement (Figure 1) compared with most other reported Adoxaceae cp genomes [12,24,25].

### 2.2. Expansion and Contraction of the Inverted Repeat Regions

The IR/single copy (SC) region junctions were analyzed across the 21 *Viburnum* and 3 *Sambucus* cp genomes (Figure 2). The *trnN-GUU*/*ndhF* and *rpl2*/*trnH-GUG* genes were detected around the IRb/SSC and IRa/LSC junction regions, respectively. The LSC/IRb junction was found to reside within the *rps19* gene, and the SSC/IRa junction was located in the *ycf1* gene. Although the boundaries of these genomic regions were highly conserved, we still observed minor differences between the two genera. At the LSC/IRb junction, except for *V. rhytidophyllum*, the IRb regions expanded by 32 bp and 116 bp toward the *rps19* gene of the remaining *Viburnum* species and *Sambucus* species, respectively. The *ndhF* gene crossed over the IRb/SSC junction in *V. cinnamomifolium* and overlapped with the IRb region by 135 bp. It was located at the SSC region in all other *Viburnum* and *Sambucus* species, and the whole length varied from 2187 bp to 2250 bp. Notably, the *ndhF* gene was found to be inverted in all Adoxaceae [26], possibly due to an early stage of the IR expansion followed by a contraction of the boundary. As for the *ycf1* gene, there were 4147–4334 bp sequences located at SSC in *Viburnum* and uniformly 4574 bp in *Sambucus*, while the fragments in IRa ranged from 1364 bp to 1547 bp in *Viburnum*, and from 1115 bp to 1126 bp in *Sambucus*. The *rpl2* gene was invariable within species in both *Viburnum* (1490 bp) and *Sambucus* (1498 bp). In addition, all the *trnH-GUG* genes within the Adoxaceae species studied here had an equal length of 75 bp except for *V. oliganthum* (78 bp). Similar IR/SC boundary structures shared among Adoxaceae species have also been reported in previous plastome studies [12,26].

### 2.3. Sequence Divergence Analysis 

To analyze the level of comprehensive sequence divergence, the 21 *Viburnum* and 3 *Sambucus* cp genome sequences were compared and plotted using the mVISTA program (See Appendix B, Figure A1). Based on the overall sequence identity, similar to most of the angiosperms, our results indicated that the LSC and SSC regions were more divergent and variable than the two IR regions [27]. In addition, the cp genomes among species in both genera showed few differences. We calculated Pi values for 213 regions in total [including 82 CDSs, 117 IGSs (intergenic spacers) and 14 introns; Figure 3]. The mean Pi values of the coding regions were 0.00418 and 0.00255, respectively, for *Viburnum* and *Sambucus*, i.e., higher than the noncoding regions (*Viburnum*: 0.0092; *Sambucus*: 0.00785), as found in the majority of angiosperms [28]. Among coding regions, the Pi values for each region ranged from 0.00012 (*rpl2*) to 0.01193 (*rps19*) in *Viburnum*, among which 10 had high values (Pi > 0.007; Table 3). In contrast, within *Sambucus*, the Pi values varied from 0.00034 (*ycf2*) to 0.00999 (*ycf1*), and the 10 most variable regions had Pi values > 0.002 (Table 3). For the 81 noncoding regions, the Pi values ranged from 0.00014 (*rpl2* intron) to 0.03129 (*psbI-trnS*) in *Viburnum*, and 0.00072 (*accD-psaI*) to 0.02934 (*trnF-ndhJ*) in *Sambucus*. The 10 most variable regions in both genera had Pi values > 0.01 (Table 3).

The chloroplast DNA region has already been used to explore the phylogenetic structure and phylogeographic patterns at different taxonomic levels. For instance, hypervariable regions of cpDNA (e.g., *matK*, *ndhF*, *rbcL*, *petB*-*petD*, *rpl32*-*trnL*, *trnC*-*ycf6*, *trnH*-*psbA*, *trnK* intron and *trnS*-*trnG*) were used to infer phylogenetic relationships for several studies with *Viburnum* [17,18,19]. Despite increased levels of confidence being revealed in most of the early branches, the relationships within clades of closely related species were still poorly resolved. Most regions used in these studies are today considered low to intermediately variable regions with low Pi values (Figure 3). Additionally, only *rpl32*-*trnL* is among the most informative regions of the plastome for most groups (Table 3). Thus, additional phylogenetically informative markers should be included to enhance the phylogenetic resolution in low-level phylogenetic or phylogeographic studies.

### 2.4. Characterization of Repeat Sequences and SSR Polymorphisms

The distribution of repetitive sequences in the cp genomes of the two genera was quite similar: the palindromic repeats were the most abundant repeat category in 16 of 21 *Viburnum* species and three *Sambucus* species, followed by forward repeats. The complement repeat was detected and occurred once only in *V. sempervirens* var. *trichophorum*, *V. melanocarpum*, *V. foetidum* var. *rectangulatum*, *V. luzonicum*, and *V. odoratissimum* var. *awabuki* (Figure 4A). On the whole, the number of both total repeats and each category of repeats (i.e., palindromic and forward repeats) in the 21 *Viburnum* species was much higher than that in the three *Sambucus* species. In all 24 plastomes, most of these repeats exhibited lengths between 30 and 59 bp, and only a minority showed long repeats, i.e., more than 60 bp in size (See Appendix B, Figure A2). In addition, the repeats were more frequently distributed in gene regions or intergenic spacer regions than in intron regions within the family Adoxaceae (See Appendix B, Figure A3). These repeat motifs have promoted the rearrangement of the cp genomes and increased the genetic diversity of populations [29], and usually are useful markers in phylogenetic analyses [30,31].

SSRs are abundantly distributed throughout the cp genome and have been widely used in species authentication and population genetics [32,33]. We found similar numbers and distribution pattern of SSR motifs among 21 and 3 accessions, respectively, in *Viburnum* and *Sambucus* (Figure 4B). The number of SSRs per plastome ranged from 47 (*V. opulus*) to 67 (*V. henryi* and *V. nervosum*) in *Viburnum*, and from 50 (*S. adnata*) to 64 (*S. nigra*) in *Sambucus*, with 119 SSRs being shared between all *Viburnum* plastomes and 149 in those of *Sambucus*. All the five kinds of SSRs (i.e., mono-, di-, tri-, tetra- and penta- nucleotide repeats) were detected in the 24 plastomes. By contrast, hexanucleotide repeats were only present in the 21 *Viburnum* species. Overall, mononucleotide SSR loci (A or T) were by far the most frequent type observed in both genera, potentially as a result of the bias toward A and T of cp genomes [34,35]. For both genera, SSRs were mainly situated in IGS (*Viburnum*: 63.77%; *Sambucus*: 67.79%; See Appendix B, Figure A3), and were also found in introns (*Viburnum*: 13.47%; *Sambucus*: 10.74%) and CDSs (*Viburnum*: 22.76%; *Sambucus*: 21.48%). These repeats will serve as useful resources for marker development for future studies on population genetics and phylogeography in Adoxaceae.

### 2.5. Phylogenetic Relationships

Based on the complete cp genome sequence dataset, two major clades were revealed, comprising a large clade and a small clade with 100% bootstrap support (Figure 5). The small clade included the genera *Sambucus*, *Adoxa*, *Tetradoxa*, and *Sinadoxa*, within which samples of *Sambucus* formed a monophyletic clade (bootstrap percentage, BS = 100%) and were sister to the *Adoxa*-*Tetradoxa*-*Sinadoxa* group. The large clade containing all *Viburnum* species was found to be monophyletic (BS = 100%) as well. Many relationships within this genus were well resolved, and the topology was almost identical to that of Clement et al. [19]. Thus, some clade names used here were taken from their study. Relationships at the base of the *Viburnum* clade were best represented by a dichotomy that included a group containing the Valvatotinus clade (represented here by *Viburnum* sect. *Viburnum*) and *Viburnum* sect. *Pseudotinus*, and a group containing all remaining *Viburnum* (Figure 5) [16,17]. In previous studies, the position of *Viburnum* sect. *Pseudotinus* was unstable. In some analyses, it (represented by *V. cordifolium*, *V. furcatum* and *V. lantanoides*) was sister to the clade with the remainder of *Viburnum* [15,16]; in other analyses, it (represented by *V. furcatum*, *V. lantanoides*, *V. nervosum* and *V. sympodiale*) appeared as sister to the Valvatotinus clade but with weak support [17,19]. However, in the present study, the sister relationship between the Valvatotinus clade and *Viburnum* sect. *Pseudotinus* was strongly supported (BS = 100%; Figure 5).

Two sister clades were clearly indicated (each 100%) within the clade that comprises all remaining *Viburnum*. The first clade Crenotinus, characterized by curving (crenate) leaf teeth [19], contained *Viburnum* sect. *Tomentosa* (represented here by *V. hanceanum*) and *Viburnum* sect. *Solenotinus*. Within the Crenotinus clade, our analysis confirmed the monophyly of the *Solenotinus* radiation (BS = 100%) and also the sister relationship between this section and *Viburnum* sect. *Tomentosa*. The second clade was Nectarotinus [19], which is characterized by extrafloral nectaries, containing the four traditionally recognized sections *Viburnum* sect. *Odontotinus*, *Viburnum* sect. *Megalotinus*, *Viburnum* sect. *Tinus*, and *Viburnum* sect. *Opulus* (represented here by *V. opulus*). Within this clade, consistent with the findings of Clement et al. [19], our analysis provided strong support for the placement of the monophyletic section *Viburnum* sect. *Tinus* as sister to the rest of the species (BS = 100%). One important difference between this result and that of Clement et al. [19] concerned the placement of *Viburnum* sect. *Opulus*. In line with previous studies [17,18], *Viburnum* sect. *Opulus* was recovered as sister to the clade containing sections *Viburnum* sect. *Odontotinus* and *Viburnum* sect. *Megalotinus* with confidence (BS = 94%). However, there was little support for this position based on the results of Clement et al. [19]. The two remaining sections, i.e., *Viburnum* sect. *Odontotinus* and *Viburnum* sect. *Megalotinus*, were clearly not monophyletic. This result was expected based on previous analyses [6,15,16,17,18,19,36]. The mostly red-fruited group of *Viburnum* sect. *Odontotinus*, namely Succodontotinus [16], was closely related to *V. cylindricum* of *Viburnum* sect. *Megalotinus*. *V. ternatum* (*Viburnum* sect. *Megalotinus*) was revealed to be sister to the polytomy consisting of the clade Succodontotinus plus *V. cylindricum* (BS = 100%).

In summary, four of the eight traditionally recognized sections in *Flora of China* were found to be monophyletic (i.e., *Viburnum* sect. *Tinus*, *Viburnum* sect. *Solenotinus*, *Viburnum* sect. *Viburnum* and *Viburnum* sect. *Pseudotinus*). The sections *Viburnum* sect. *Odontotinus* and *Viburnum* sect. *Megalotinus* were recovered as nonmonophyletic, which has been repeatedly shown in various molecular and morphological analyses [6,15,16,17,18,19,36]. Only a single representative was included in our analyses, for sections *Viburnum* sect. *Tomentosa* and *Viburnum* sect. *Opulus*. Additional sampling will be required to evaluate the monophyly of these groups. In addition, many relationships within our 45-species plastid tree were confidently resolved and the topology was identical to that of Clement et al. [19], with the exception of the relationships between the Valvatotinus clade and *Viburnum* sect. *Pseudotinus* and the position of *V. opulus* (*Viburnum* sect. *Opulus*). In the first case, there was strong support for the clade Valvatotinus being sister to *Viburnum* sect. *Pseudotinus* (100% bootstrap value; Figure 5). In the other case, as expected, *V. opulus* of *Viburnum* sect. *Opulus* was found to be sister to the clade comprising sections of *Viburnum* sect. *Odontotinus* and *Viburnum* sect. *Megalotinus*, which generally maintained its previously determined position in relation to *Viburnum* sect. *Megalotinus* and *Viburnum* sect. *Odontotinus*.

## 3. Materials and Methods

### 3.1. Sample Collection, Sequencing and Assembly

Fresh leaves from 21 species of *Viburnum*, representing all of the 8 sections recognized in *Flora of China*, together with 3 species of *Sambucus*, were sampled in China (Appendix A) and dried in silica gel. The voucher specimens were deposited in College of Plant Protection, Henan Agricultural University (Appendix A). Total genomic DNA of the 24 species was extracted and then sequenced on an Illumina Hiseq2500 Platform at Jinweizhi Biotechnology Institute (Suzhou, China).

We used a combination of de novo and reference-guided methods to assemble these plastomes [37]. Firstly, for each *Viburnum* and *Sambucus* species, raw paired-end reads were trimmed to remove low-quality reads with a Phred value < 20 using CLC Genomics Workbench v10.1.1 (CLC Bio, Aarhus, Denmark; http://www.clcbio.com). Secondly, the remaining clean reads were assembled into contigs on the CLC assembler with the following settings: bubble size, 98; minimum contig length, 250 bp; mismatch cost, 2; deletion and insertion costs, 3; length fraction, 0.9; and similarity fraction, 0.8. Thirdly, due to the fact that the original sequences represented a mixture of both nuclear and organellar DNA, to filter the plastid-like ones, all contigs of *Viburnum* and *Sambucus* were aligned to the reference genomes *Viburnum betulifolium* (GenBank accession number: NC_037951) and *Sambucus williamsii* (GenBank accession number: NC_033878), respectively, using BLAST (http://blast.ncbi.nlm.nih.gov/). Then, the filtered contigs longer than 10 kb were oriented and realigned with the reference genomes for constructing the draft chloroplast genome of each species with GENEIOUS V11.01 software (http: //www.geneious.com). Finally, the ordered contigs were remapped to the draft genome to generate the complete chloroplast genome sequences. To validate the assembly, PCR amplifications and Sanger sequencing were performed to confirm the four junction regions between IRs and LSC/SSC with primers developed from assembled sequences flanking the junction regions (Appendix A).

### 3.2. Whole Chloroplast Genome Annotation and Comparison

The whole chloroplast genomes were annotated using GENEIOUS V11.01 and DOGMA [38]. The start/stop codons and intron/exon boundaries of genes were checked and adjusted manually according to the reference genomes. In addition, the tRNA boundaries were further verified by tRNAscan-SE v1.21 [39] with default settings. Online program OrganellarGenome DRAW (https://chlorobox.mpimp-golm.mpg.de/OGDraw.html) [40] was used to draw the gene maps of *Viburnum* and *Sambucus* cp genomes. Finally, the 24 annotated plastome sequences were deposited in GenBank.

Chloroplast genome comparisons across the 21 *Viburnum* and 3 *Sambucus* species were conducted on the mVISTA tool (genome.lbl.gov/vista/index.shtml) [41] using Shuffle-LAGAN mode, with the annotations of *V. betulifolium* and *S. williamsii* serving as references, respectively. In order to identify the variant hotspot regions for *Viburnum* and *Sambucus*, the sequence alignments of their respective plastomes were subjected to a sliding window analysis in DNASP v5.10 [42] to estimate the nucleotide variability (Pi) for all the protein coding and noncoding regions (i.e., IGSs and introns).

### 3.3. Identification of Repeat Sequences and SSRs

The whole cp genomes of *Viburnum* and *Sambucus* were aligned in GENEIOUS v11.1.4 using MAFFT multiple aligner v7 [43], respectively. Then, chloroplast SSR loci (i.e., mono-, di-, tri-, tetra-, penta- and hexa- nucleotide repeats) were identified using Perl script MISA (http://pgrc.ipk-gatersleben.de/misa/misa.html) with minimal repeat numbers of 10, 5, 4, 3, 3 and 3 for mononucleotide, dinucleotide, trinucleotide, tetranucleotide, pentanucleotide and hexanucleotide repeats, respectively. Moreover, the program REPUTER [44] was used to estimate the number and position of repeat elements, including direct (forward), inverted (palindromic), complement and reverse repeats. The constraints to all the four repeat types in REPUTER were 1) a minimum repeat size of 30 bp; and 2) 90% higher sequence identity with a hamming distance of 3 (i.e., the maximum length of the gap size between repeats equals 3 bp).

### 3.4. Phylogenetic Analysis

We used 45 cp genomes to infer the phylogenetic relationships among Adoxaceae species, including 24 newly obtained plastomes, 19 plastomes downloaded from the GenBank (i.e., 6 plastomes of *Viburnum* sect. *Odontotinus*, 1 plastome of *Viburnum* sect. *Megalotinus*, 3 plastomes of *Viburnum* sect. *Solenotinus*, 2 plastomes of *Viburnum* sect. *Viburnum*, plus 7 representatives of *Sambucus*, *Adoxa*, *Sinadoxa*, and *Tetradoxa*) and two outgroups, *Panax ginseng* and *Eleutherococcus nodiflorus* (Appendix A). The phylogenetic analysis was performed with a maximum-likelihood (ML) method based on the complete cp genome sequence dataset. Chloroplast sequences of these 45 species were aligned together using MAFFT with default settings. ML analysis was conducted in RAXML-HPC [45] on the CIPRES cluster (http://www.phylo.org/), with a GTR+G+I substitution model selected by jModelTest v2.1.7 [46] and an unpartitioned strategy.

## 4. Conclusions

This work presents a major advance in understanding Chinese Adoxaceae phylogenetics and plastome evolution with a particular focus on the genus *Viburnum*. The comparison of the plastomes among each species of *Viburnum* and *Sambucus*, and with those of other members of Adoxaceae, revealed high similarities with respect to genomic structure, gene order and content. Repeat sequences, SSRs and highly variable regions were identified with the purpose of developing potential molecular markers for future studies on the population genetics, phylogeny and phylogeography of *Viburnum* and *Sambucus*. Our phylogenomic analysis, based on the complete cp genome sequence dataset, strongly supported the relationships within Adoxaceae and *Viburnum* revealed by previous plastid phylogenomic investigations [12,19,25,26]. *Viburnum* was shown to be a sister to the *Sambucus-Adoxa-Tetradoxa-Sinadoxa* group. Within *Viburnum*, the monophyly of four traditionally recognized sections in *Flora of China* (i.e., *Viburnum* sect. *Tinus*, *Viburnum* sect. *Solenotinus*, *Viburnum* sect. *Viburnum* and *Viburnum* sect. *Pseudotinus*) was strongly supported. The nonmonophyly of sections *Viburnum* sect. *Odontotinus* and *Viburnum* sect. *Megalotinus* was repeatedly demonstrated. Additionally, our analyses confirmed the sister relationships between the clade Valvatotinus and *Viburnum* sect. *Pseudotinus*, as well as between *Viburnum* sect. *Opulus* and the *Odontotinus-Megalotinus* group. Overall, our results clearly exhibited the power of the complete cp genomes to improve phylogenetic resolution, and will contribute to a better understanding of plastome evolution in Chinese Adoxaceae.

## Figures and Tables

**Figure 1 plants-09-01143-f001:**
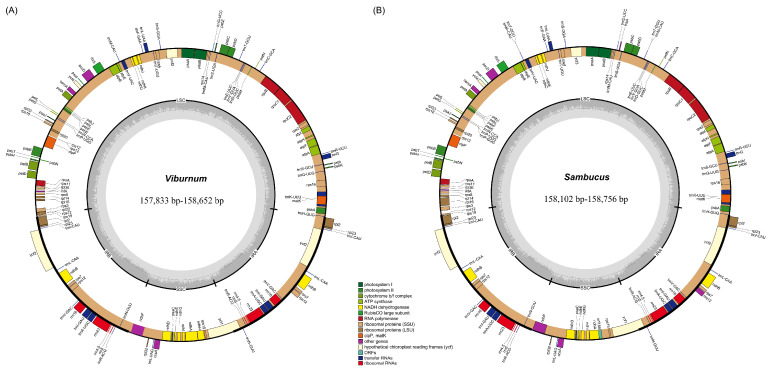
Chloroplast genome maps for (**A**) 21 *Viburnum* species and (**B**) 3 *Sambucus* species.

**Figure 2 plants-09-01143-f002:**
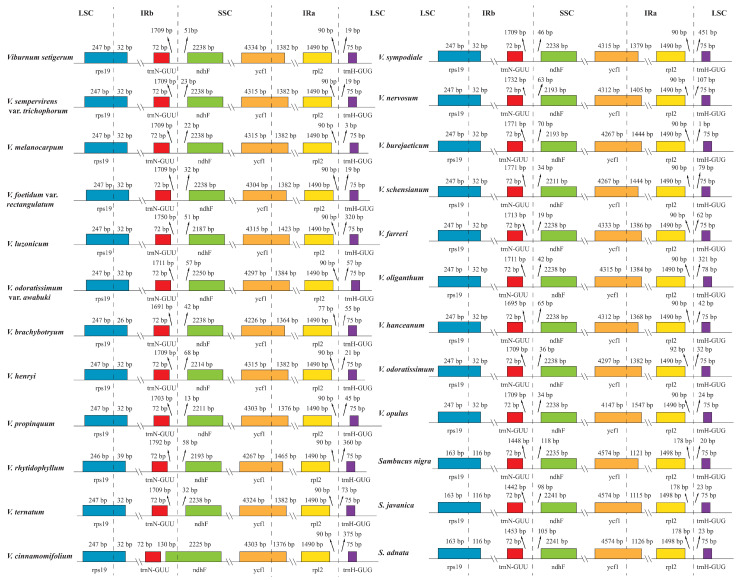
Comparison of the junctions between IRs and SSC/LSC regions for the 24 Adoxaceae species.

**Figure 3 plants-09-01143-f003:**
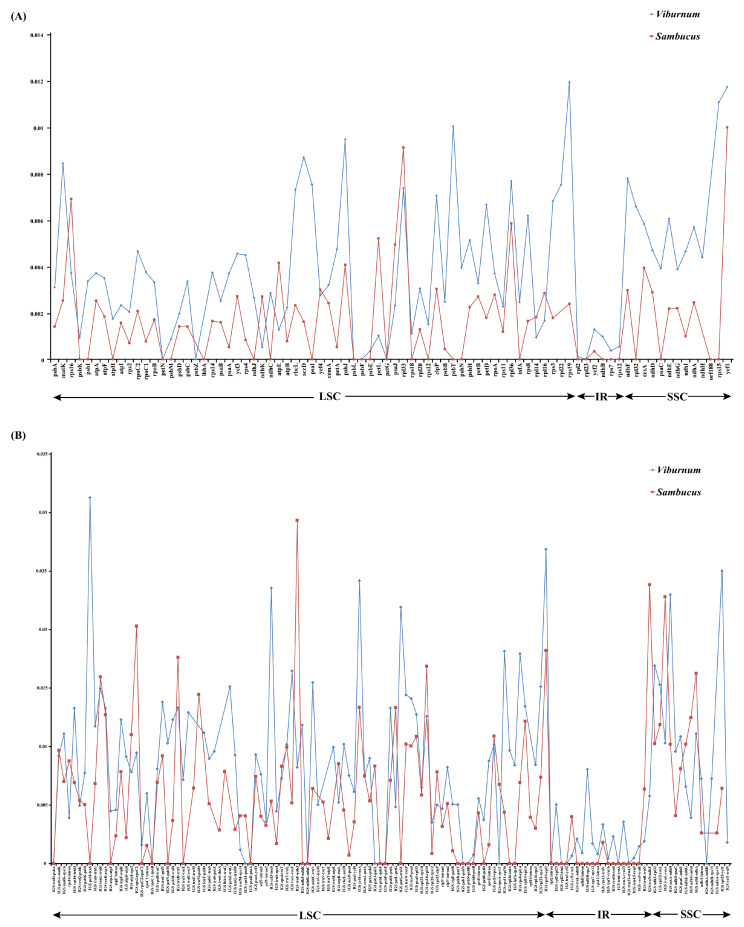
Percentages of variable characteristics in homologous regions among the chloroplast genomes of 24 Adoxaceae species. (**A**) Pi values among CDSs. (**B**) Pi values of intergenic spacer (IGS) regions and introns.

**Figure 4 plants-09-01143-f004:**
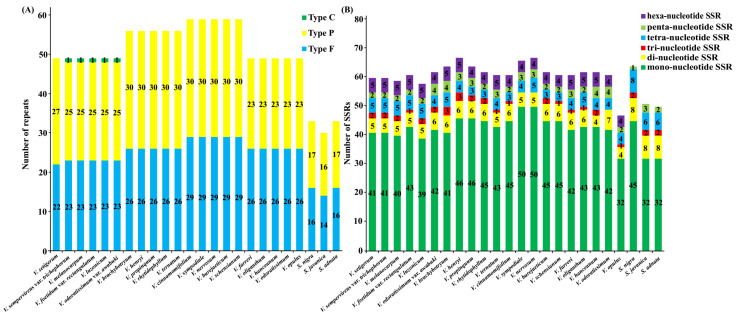
The distribution of repeats and SSRs in the chloroplast genomes of 24 Adoxaceae species. (**A**) Frequency of repeat types in 24 Adoxaceae species. F, P and C indicate the forward, palindrome and complement repeat types, respectively. (**B**) Compositions of the SSR in 24 Adoxaceae species. Different SSR motifs are shown in different colors.

**Figure 5 plants-09-01143-f005:**
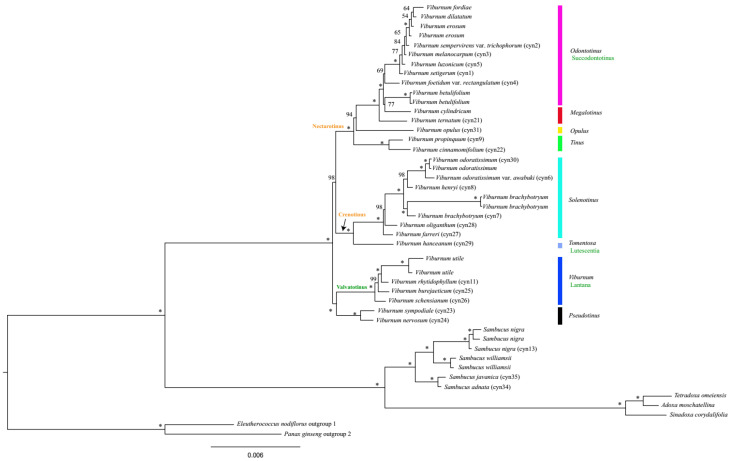
Phylogenetic relationships among *Viburnum* and *Sambucus* species inferred from the complete cp genome sequence dataset based on maximum likelihood (ML) method. Numbers above the nodes represent ML bootstrap values, and “*” indicates 100% support values in ML. Clade names (in the right) marked in black represent previously published names under the ICN (International Code of Nomenclature) that are here converted to phylogenetic names. Those in green (in the right and along a branch) represent names proposed by Winkworth and Donoghue [16] and Clement and Donoghue [17]. Clade names in orange (along a branch) represent names proposed by Clement et al. [19].

**Table 1 plants-09-01143-t001:** Summary of the main characteristics of Adoxaceae plastomes.

Species	Genome Size (bp)	LSC Length (bp)	SSC Length (bp)	IR Length (bp)	Total GC Content (%)	Number of Genes			
						Total	CDS	rRNAs	tRNAs
*V. setigerum*	158,306	86,763	18,539	26,502	38.1%	130	85 (6)	8 (4)	37 (7)
*V. sempervirens* var. *trichophorum*	158,184	86,710	18,472	26,501	38.1%	130	85 (6)	8 (4)	37 (7)
*V. melanocarpum*	158,196	86,695	18,497	26,502	38.1%	130	85 (6)	8 (4)	37 (7)
*V. foetidum* var. *rectangulatum*	158,230	86,835	18,431	26,482	38.1%	130	85 (6)	8 (4)	37 (7)
*V. luzonicum*	158,652	87,892	17,674	26,543	38.1%	130	85 (6)	8 (4)	37 (7)
*V. odoratissimum* var. *awabuki*	158,126	86,718	18,438	26,485	38.1%	130	85 (6)	8 (4)	37 (7)
*V. brachybotryum*	157,833	86,809	18,268	26,378	38.1%	130	85 (6)	8 (4)	37 (7)
*V. henryi*	157,862	86,430	18,452	26,490	38.1%	130	85 (6)	8 (4)	37 (7)
*V. propinquum*	157,987	86,839	18,350	26,399	38.1%	130	85 (6)	8 (4)	37 (7)
*V. rhytidophyllum*	158,520	87,054	18,338	26,564	38.1%	130	85 (6)	8 (4)	37 (7)
*V. ternatum*	158,344	87,109	18,407	26,414	38.1%	130	85 (6)	8 (4)	37 (7)
*V. cinnamomifolium*	158,347	87,210	18,347	26,395	38.1%	130	85 (6)	8 (4)	37 (7)
*V. sympodiale*	158,238	87,118	18,330	26,395	38.0%	130	85 (6)	8 (4)	37 (7)
*V. nervosum*	157,890	86,715	18,341	26,417	38.0%	130	85 (6)	8 (4)	37 (7)
*V. burejaeticum*	157,913	86,669	18,274	26,485	38.1%	130	85 (6)	8 (4)	37 (7)
*V. schensianum*	157,924	86,681	18,289	26,477	38.1%	130	85 (6)	8 (4)	37 (7)
*V. farreri*	158,046	86,809	18,401	26,418	38.1%	130	85 (6)	8 (4)	37 (7)
*V. oliganthum*	158,309	87,038	18,453	26,409	38.1%	130	85 (6)	8 (4)	37 (7)
*V. hanceanum*	158,195	86,815	18,436	26,472	38.1%	130	85 (6)	8 (4)	37 (7)
*V. odoratissimum*	158,020	86,653	18,419	26,474	38.1%	130	85 (6)	8 (4)	37 (7)
*V. opulus*	158,520	87,114	18,456	26,475	38.2%	130	85 (6)	8 (4)	37 (7)
*S. nigra*	158,102	86,518	18,978	26,303	38.0%	132	84 (6)	8 (4)	40 (7)
*S. javanica*	158,624	87,226	18,854	26,272	38.0%	132	84 (6)	8 (4)	40 (7)
*S. adnata*	158,756	87,328	18,862	26,283	38.0%	132	84 (6)	8 (4)	40 (7)

Numbers in brackets indicate the numbers of genes duplicated in the IR regions.

**Table 2 plants-09-01143-t002:** Gene composition in the 24 Adoxaceae chloroplast genomes.

Gene Group	Gene Name
Ribosomal RNAs	*rrn16*(×2), *rrn23*(×2), *rrn4.5*(×2), *rrn5*(×2)
Transfer RNAs	*trnH-GUG*, *trnK-UUU*^a^, *trnQ-UUG*, *trnS-GCU*, *trnG*^a^, *trnR-UCU*
	*trnC-GCA*, *trnD-GUC*, *trnY-GUA*, *trnE-UUC*, *trnT-GGU*
	*trnS-UGA*, *trnG-UCC*, *trnfM-CAU*, *trnS-GGA*, *trnT-UGU*
	*trnL-UAA*^a^, *trnF-GAA*, *trnV-UAC*^a^, *trnM-CAU*, *trnW-CCA*
	*trnP-UGG*, *trnl-CAU*(×2), *trnL-CAA*(×2), *trnV-GAC*(×2)
	*trnl-GAU*^a^(×2), *trnA-UGC*^a^(×2), *trnR-ACG*(×2), *trnN-GUU*(×2)
	*trnL-UAG*, *trnM-CAU*_x_, *trnT-GGU*_x_, *trnP-GGG*_x_
Photosystem I	*psaB*, *psaA*, *psal*, *psaJ*, *psaC*
Photosystem II	*psbA*, *psbK*, *psbl*, *psbM*, *psbD*, *psbC*, *psbZ*_z_, *psbB*, *psbT*,
	*psbL*, *psbF*, *psbE*, *psbH*, *psbN, psbJ*
Cytochrome	*petN*, *petA*, *petL*, *petG*, *petB*^a^, *petD*^a^
ATP synthase	*atpA*, *atpF*^a^, *atpH*, *atpl*, *atpE*, *atpB*
Rubisco	*rbcL*
NADH dehydrogenease	*ndhJ*, *ndhK*, *ndhC*, *ndhB*^a^(×2), *ndhD*, *ndhE*, *ndhG*
	*ndhl*, *ndhA*^a^, *ndhH*_z_
Ribosomal proteins (large units)	*rpl33*, *rpl20*, *rpl36*, *rpl14*, *rpl16*, *rpl16*^a^, *rpl22*_z_, *rpl2*^a^(×2),
	*rpl23*(×2), *rpl32*
Ribosomal proteins (small units)	*rps16*^a^, *rps2*, *rps14*, *rps4*, *rps18*, *rps12*^b^ (×2), *rps11*, *rps8*,
	*rps7*(×2), *rps15*, *rps3, rps19*
RNA polymerase	*rpoC2*, *rpoC1*^a^, *rpoB*, *rpoA*
Miscellaneous proteins & ATP-dependent protease subunit P	*matK*, *clpP*^b^
Other genes	*accD*, *cemA*, *infA*, *ccsA*, *orf188*_x_, *lhbA*_x_
Hypothetical proteins & Conserved reading frame	*ycf3*^b^, *ycf4*, *ycf2*(×2), *ycf1*_Ψ_

^a^ Indicates the genes containing a single intron. ^b^ Indicates the genes containing two introns. _x_ Indicates the gene is present only in Sambucus. _z_ Indicates the gene is present only in Viburnum. (×2) indicates genes duplicated in the IR regions; pseudogene is represented by _Ψ_. ycf1 is a pseudogene only in the following 16 species: *V. setigerum*, *V. foetidum* var. *rectangulatum*, *V. luzonicum*, *V. odoratissimum* var. *awabuki*, *V. brachybotryum*, *V. rhytidophyllum*, *V. sympodiale*, *V. nervosum*, *V. burejaeticum*, *V. schensianum*, *V. farreri*, *V. oliganthum*, *V. hanceanum*, *S. nigra*, *S. javanica*, *S. adnata*.

**Table 3 plants-09-01143-t003:** Pi values of the ten most variable coding and noncoding regions in *Viburnum* and *Sambucus*.

*Viburnum*	Pi	*Sambucus*	Pi
noncoding regions			
*rps15-ycf1*	0.02502	*trnF-ndhJ*	0.02934
*ycf4-cemA*	0.02419	*trnN-ndhF*	0.02384
*ycf3-trnS*	0.02356	*rps2-rpoC2*	0.02029
*ccsA-ndhD*	0.02298	*rps18-rpl20*	0.01687
*rps8-rpl14*	0.01794	*trnG-trnR*	0.01596
*ndhF-rpl32*	0.01691	*trnM-psbD*	0.01446
*trnL-trnF*	0.01647	*ycf4-cemA*	0.01335
*ndhC-trnV*	0.01548	*ndhG-ndhI*	0.01247
*rpl32-trnL*	0.01529	*rpl32-trnL*	0.01188
*psbZ-trnG*	0.01512	*atpI-rps2*	0.01101
coding regions			
*rps19*	0.01193	*ycf1*	0.00999
*ycf1*	0.01173	*rpl33*	0.00912
*rps15*	0.01107	*rps16*	0.00691
*accD*	0.00870	*atpE*	0.00415
*matK*	0.00844	*ccsA*	0.00394
*ndhF*	0.00779	*clpP*	0.00303
*rpl22*	0.00753	*ycf4*	0.00300
*rpl33*	0.00737	*ndhF*	0.00298
*rbcL*	0.00730	*ndhD*	0.00288
*clpP*	0.00705	*rpl16*	0.00285

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
