# Peer review of "Phylogenetic and Comparative Analyses of Complete Chloroplast Genomes of Chinese Viburnum and Sambucus (Adoxaceae)"

_plants, 2020, doi:10.3390/plants9091143_

Round 1

Reviewer 1 Report

The manuscript entitled “Phylogenetic and Comparative Analyses of Complete Chloroplast Genomes of Chinese Viburnum and Sambucus (Adoxaceae)” presents results of study on complete genomes of woody plants from Adoxaceae family: 21 newly sequenced species of Viburnum and three representatives of Sambucus, native to China. The species of Viburnum genus represent all the eight currently diagnosed sections in Flora of China. The aims of this study were: i) characterization and comparison of plastid genomes with a focus on evolutionary patterns, ii) phylogenetic inference and estimation of molecular divergence dates, iii) identification of repeat sequences, simple sequence repeats and mutational hotspots regions. All these goals have been achieved through research reported in the manuscript. The Authors thoroughly characterized the structure and sequence variation within the genus Viburnum and Sambucus and discussed it against the data on other representatives of Adoxaceae.

Study presented in the manuscript is well planned and well done. The analyzes are multi-threaded and exhaustive. The research is clearly and concisely described. English is understandable and requires only minor corrections. I positively evaluate the verification of the correctness of the assembly of genomes (points of contact between IR and SSC or LSC) by PCR amplification ad Sanger sequencing. The strong point of the article is also the enrichment of the phylogenetic analysis by dating the diversification events on a phylogenetic tree.

I highly rate the manuscript submitted for my review. The results and conclusions presented in the paper surely will make an important contribution to the knowledge in the context of phylogenetic relationships within Adoxaceae, as well as organization and variability of plant chloroplast genome in general.

Below detailed comments and questions are listed:

Line 16: species that endemic to China – should be species endemic to China

Line 40: used as folk medicine – should be used in folk medicine.

Line 62: Double space. (Please, check all the text for double spaces)

The caption under Table 2: Indicates the gene that present only ­– should be Indicates the gene present only (the wording used twice in a line).

Line 161-162: I would advise to sort PIC values in descending order in both parentheses. I would also recommend developing shortcuts like PICs when they first appear in the text, not only in the chapter Materials and Methods.

Line 212: Shouldn’t You probably write: Sambucus, Adoxa, Tetradoxa and Sinadoxa? Also, in line 214 the name of a group is Adoxa-Tetradoxa-Tetradoxa. Why not -Sinadoxa? Please, correct or explain because it is confusing.

Figure 5: In a clade comprising Tetradoxa, Adoxa and Sinadoxa there are 100% support values given over the branches. Why not “*”?

Line 251: Similar situation with the name of a clade like in line 214.

Line 270: I would recommend writing a full wording in a caption instead of HPD shortcut only.

Line 345: The word represents should, in my opinion, be changed to presents.

Line 352: The word that should be deleted form the sentence. Previous – no capital letter is needed.

Line 356: since the end of Eocene – please verify the use of word since in this case.

Line 357: Please, delete the word “was” from the sentence.

Author Response

Manuscript ID: plants-876632

Title: Phylogenetic and Comparative Analyses of Complete Chloroplast Genomes of Chinese Viburnum and Sambucus (Adoxaceae)

COMMENTS TO AUTHOR

SPECIFIC RESPONSES TO Reviewers’ comments (All line numbers mentioned below refer to the revised clean version):

Reviewer: 1

Comments to Author

The manuscript entitled “Phylogenetic and Comparative Analyses of Complete Chloroplast Genomes of Chinese Viburnum and Sambucus (Adoxaceae)” presents results of study on complete genomes of woody plants from Adoxaceae family: 21 newly sequenced species of Viburnum and three representatives of Sambucus, native to China. The species of Viburnum genus represent all the eight currently diagnosed sections in Flora of China. The aims of this study were: i) characterization and comparison of plastid genomes with a focus on evolutionary patterns, ii) phylogenetic inference and estimation of molecular divergence dates, iii) identification of repeat sequences, simple sequence repeats and mutational hotspots regions. All these goals have been achieved through research reported in the manuscript. The Authors thoroughly characterized the structure and sequence variation within the genus Viburnum and Sambucus and discussed it against the data on other representatives of Adoxaceae.

Study presented in the manuscript is well planned and well done. The analyzes are multi-threaded and exhaustive. The research is clearly and concisely described. English is understandable and requires only minor corrections. I positively evaluate the verification of the correctness of the assembly of genomes (points of contact between IR and SSC or LSC) by PCR amplification ad Sanger sequencing. The strong point of the article is also the enrichment of the phylogenetic analysis by dating the diversification events on a phylogenetic tree.

I highly rate the manuscript submitted for my review. The results and conclusions presented in the paper surely will make an important contribution to the knowledge in the context of phylogenetic relationships within Adoxaceae, as well as organization and variability of plant chloroplast genome in general.

Response: Thank you for your thoughtful and detailed comments and recommendations. We believe we have been able to incorporate all of these suggestions into the revised version of our manuscript.

Suggestions:

#1) Line 16: species that endemic to China – should be species endemic to China

Response: Done (Line 16).

#2) Line 40: used as folk medicine – should be used in folk medicine.

Response: Done (Line 49).

#3) Line 62: Double space. (Please, check all the text for double spaces)

Response: Done (Line 70).

#4) The caption under Table 2: Indicates the gene that present only ­– should be Indicates the gene present only (the wording used twice in a line).

Response: Done. Please see the new footnote under Table 2.

#5) Lines 161-162: I would advise to sort PIC values in descending order in both parentheses. I would also recommend developing shortcuts like PICs when they first appear in the text, not only in the chapter Materials and Methods.

Response: We have rephrased the first paragraph of “2.3. Sequence Divergence Analysis” (Lines 198-271) and removed the PIC analysis and related results from the text based on suggestions of Reviewer 2.

#6) Line 212: Shouldn’t You probably write: Sambucus, Adoxa, Tetradoxa and Sinadoxa? Also, in line 214 the name of a group is Adoxa-Tetradoxa-Tetradoxa. Why not -Sinadoxa? Please, correct or explain because it is confusing.

Response: Done. Indeed, these were genuine mistakes, we have changed Tetradoxa in lines 361, 363 to Sinadoxa.

#7) Figure 5: In a clade comprising Tetradoxa, Adoxa and Sinadoxa there are 100% support values given over the branches. Why not “*”?

Response: Done. These were genuine oversights, and we have changed 100% over the branches comprising Tetradoxa, Adoxa and Sinadoxa to “*” in Figure 5.

#8) Line 251: Similar situation with the name of a clade like in line 214.

Response: We have removed the time-calibration analysis and related sentences throughout the text.

#9) Line 270: I would recommend writing a full wording in a caption instead of HPD shortcut only.

Response: Following the referee’s suggestion (i.e., Reviewer 2), we have removed the time-calibration analysis and related sentences in Abstract, Results and Discussion, and Conclusions, also, the Figure 6 and Table 3 from the text.

#10) Line 345: The word represents should, in my opinion, be changed to presents.

Response: Done. We have changed the word ‘represents’ to ‘presents’ (Line 521).

#11) Line 352: The word that should be deleted form the sentence. Previous – no capital letter is needed.

Response: Done (Line 529).

#12) Line 356: since the end of Eocene – please verify the use of word since in this case.

Response: We have removed the time-calibration analysis and related sentences from the manuscript throughout the text.

#13) Line 357: Please, delete the word “was” from the sentence.

Response: We have removed the time-calibration analysis and related sentences from the manuscript throughout the text.

Reviewer 2 Report

Dear Editor and Authors,

The manuscript presents a careful analysis of more than 20 chloroplast genomes of Viburnumand Sambucus, with full descriptions of the genomic structure. A plastid phylogeny focusing mostly in Viburnumis also presented, as well as a chronogram. The paper is original and has inherent value, but it is not possible to accept it as is, therefore I am recommending Reconsider after major revision.

Even though the chloroplast descriptions are good, I have objections about the trees presented in the paper, and highly recommend that the results describing them to be rewritten. Several terms related to phylogenetics are misused and do need to be corrected. Also, the time-calibration analysis presents serious conceptual flaws, and I recommend it to be removed from the paper, as it does not contribute with relevant results/discussion. I present some of the arguments below and also some references to be consulted.

I think the methodology that the authors used to assemble the chloroplasts very unusual, as they used tools (BLAST) that are not traditionally used for small genome assembly. I worry highly repetitive regions might have been thrown away during the de novo assembly. Having a close reference genome, the authors could have easily used an assembler like BWA or Bowtie2, that are optimized to deal with short-read sequences, or even combined methodologies, such as Fast-Plast. Given the very short N50 found for all samples, I’d like to see a comparison of the authors’ methods and a reference assembly, to confirm the results. I also would like to request that the authors include a column in their Supplemental Material Table 1 indicating the sequencing depth (how many times each position in the genome was sequenced), as this is a good indication of the assembly quality.

I think the conclusion could be better worked. The claim that this is a major advance in Adoxaceae phylogeny is fickle, as the paper presents a moderately-sampled phylogeny of only one large genus in the family. Also, the authors state that the plastomes were compared with other members of Adoxaceae, but this is not shown in the manuscript at any point.

The manuscript would also benefit of a review of the English language and style, as I found several parts of the text to be confusing or misleading. I’ve pointed out several of them below, but there are more that I haven’t listed.

Please see the point-by-point comments below, the numbers indicate line numbers.

Line 31: clarify what you mean by large and small genera. Just adding a parenthesis saying “small genera (less than 20 species)”, for example, would be enough.

33: it isn’t possible to say this is a clade without having a phylogeny showing it is a monophyletic lineage. Please either include a reference for a phylogeny of the genus, or just use “group” or “genus”.

36, 60, etc: I think the use of Latin America to delimit geographic distribution is very weird, given this is a delimitation based on language/culture and not geography, besides the fact that the current definition of Latin America excludes a few countries in Central and South America and the Caribbean that had French/English/Dutch colonization or that are parts of the French territory. Using Central and South America would be more appropriate, please correct throughout the manuscript.

37: change “occurred” to “occurring”

51: change “studied species” to “species sampled in phylogenetic studies” or similar, to make the sentence clearer

62: double space between taxa and richness

61 to 65: please rewrite the whole sentence that starts with “By contrast….”. It’s very convoluted and hard to follow, and I couldn’t understand if the relationships of Viburnum were not well studied in China or South/Central America and Southeast Asia.

68: rewrite objective 1. Suggestion: “characterize and compare the cp genomes of Viburnum species belonging to all the eight sessions occurring in China and related taxa in order to gain insights into their evolutionary patterns;”

80: is this number correct? “with N50 contigs varying from 285 to 399 bp”. These contigs look very short for a chloroplast assembly. Subsequently the authors state “The four junctions between IRs and SSC/LSC in each 82 species were initially determined on the basis of these contigs”. If the contigs were that small, how did you identify junctions based on them?

83: the language in this sentence is misleading. What do the authors mean with “the relevant results”? Does it mean there were results that disagreed with what is being shown in the paper?

Table 1: the footnote says “numbers in brackets” while the table only contain parentheses. Include authority names for the species listed in the table.

Table 2:

Header: correct table header to “Gene group” and “Gene name”

Footnote: correct “Indicates the gene that present only in Sambucus. Indicates the gene that present only in Viburnum.” to something like: “Indicates the gene is present only in Sambucus. Indicates the gene is present only in Viburnum.”

Footnote: correct “pseudogene was represented by Ψ.” to “pseudogene is represented by Ψ.”

94: remove the “that” and correct: “including a pair of IR regions (26,272–26,564 bp) separating the LSC region (86,430–87,892 bp) and the SSC region (17,674–18,978 bp).”

100: not possible to understand this sentence: “Notably, five genes (i.e., trnM-CAU, trnT-GGU, trnP-GGG, orf188, and lhbA) and three genes (i.e., psbZ, ndhH, and rpl22) were, respectively, presented in Sambucus and Viburnum.” Do the authors mean genes that occur exclusively in one species or other?

104: simplify this sentence: “In particular, the rps12 was trans-spliced gene, with the first exon located in the LSC region, and the second and third in the IR regions.”

106: correct: “We also found that the ycf1 gene at the SSC and IRa border was present as a pseudogene in 16 Viburnum species (Table 2), due to incomplete gene duplication, as shown by previous reports [20, 21].”

112: move the “herein” to right after “obtained”.

115: there’s absolutely no basis for this affirmation: “This is expected given the similarities of morphological characters among them.” There is no direct correlation between the similarities in chloroplast structure to morphology. One good example is family Asteraceae, the family has more than 25 thousand species and the plastomes usually have less than 1% divergence between them. Please remove.

122: correct: “the IRb/SSC and IRa/LSC junction regions,”

122: decide between using junction or border. Junction is more appropriate.

124: This sentence is misleading: “Although the boundaries of these species were highly conserved”. Change species to “genomic regions” or similar.

125, 126: simplify this sentence

127: split this sentence in two: “The ndhF gene crossed over the IRb/SSC junction in Viburnum cinnamomifolium and overlapped with the IRb region by 135 bp. It was located at the SSC region in all other Viburnum and Sambucus species, and the whole length varied from 2187 bp to 2250 bp.”

130: correct: “Notably, the ndhF gene was found to be inverted in all Adoxaceae [26], possibly due to an early stage of the IR expansion followed by a contraction of the boundary.”

133: This doesn’t make sense: “The rpl2 gene exhibited invariable sequences among intragenus species in both Viburnum (1490 bp) and Sambucus (1498 bp).” All species are intragenus (?), besides, it’s hard to understand if the whole gene was totally invariable or just parts of it. Correct to something like: “The rpl2 gene was invariable within species in both Viburnum (1490 bp) and Sambucus (1498 bp).”

135: correct the sentence: “In addition, all the trnH-GUG genes within the Adoxaceae species studied here had an equal length of 75 bp except for Viburnum oliganthum (78 bp).”

136: Phenomenon is a weird word here, and the sentence is convoluted. Change for something like this: “Similar IR/SC boundary structures shared among genera has also been reported in previous plastome studies [12, 26].”

145: change “like” to “similarly to most…”

143-165: this whole paragraph could be summarized as a table. Give the Pi values for the genes in both genera in a table and just cite this table in the text. You can reduce this paragraph and give only the highest and lowest values for each genus.

166-175: there are several things that need to be changed in this paragraph. First, it’s quite confusing trying to understand what the authors mean by “interspecies level” and “species set”. In the first sentence, I’d say to change it to “different taxonomic levels”. In the second case, just mention the markers were used in several studies with Viburnum. The paragraph also contain long sentences, that could be edited.

The most serious thing, though, is the confusion in the meaning of deep and shallow levels of the phylogeny. Traditionally these words are used to distinguish between the backbone and the tips of the tree, with “deep” being used with the meaning of “deeper in time”. Therefore, deep = closer to the backbone, shallow = closer to the tips. This confusion in the meaning is also found in other parts of the manuscript, such as line 23 in the abstract, and 55-58 in the Introduction.

I suggest the authors consult different papers about phylogeny to clarify and understand better this issue, such as https://link.springer.com/article/10.1186/s12862-016-0769-y,https://onlinelibrary.wiley.com/doi/abs/10.2307/1223970,https://onlinelibrary.wiley.com/doi/abs/10.1111/j.1365-3113.2006.00355.x?casa_token=yiMTxaqNU0MAAAAA:aCKAp-NggwaPkbLYkFEnx-aOUqro_HgFZm7HhJJhVunfqnQRFL3EK3Cvk1q7AzQ0Nr3ZZLH2tOcr4HM.

Please correct.

179: Don’t use “Sambucus plants”, use “Sambucus species”. Correct in the whole manuscript.

211: correct “supports” to “support”

212-214: The two clades in the small clade are reciprocal sister-clades, please correct the sentence.

210-238: I have several issues with this paragraph. Given the way the phylogeny is described, I have the impression the authors are not familiar with phylogenetic analysis and the jargon involved. One example is the use of “basal”. This has been a long debate in systematics, given that the term is misinterpreted most of the times (see examples: https://pubmed.ncbi.nlm.nih.gov/16701355/,https://onlinelibrary.wiley.com/doi/full/10.1111/j.0307-6970.2004.00262.x). I don’t understand the need to name two subclades, when you could refer directly to the section names.

One serious issue here is the incorrect use of taxonomic nomenclature. Sections should ALWAYS be written with the correspondent generic name, they are part of the binomial. E.g.: Viburnumsect. Tomentosa,Viburnumsect. Tinus, etc. Please refer to the INC for details (https://www.iapt-taxon.org/nomen/main.php). This needs to be corrected in the figure 5 too. When using informal names, such as clades Sambucina, etc., usually these names are not italicized, as they don’t have a formal rank.  

This whole paragraph needs to be rewritten, taking into account the explained above, and also to simplify the text. As it is right now, it’s a wall of text, without any breaks. Please re-read and reformulate, mainly using shorter sentences and maybe splitting the paragraph in two.

Figure 6 and 246-267: I don’t understand the need to include a time-calibrated tree in this study, especially with the conceptual problems present in the analysis. The dating conducted in this study is highly questionable given the choice of outgroups and calibration point, and I strongly recommend completely removing this part from the paper.

The choice of outgroup and ingroup sampling in a phylogeny can highly impact and bias results of comparative analysis such as calibration, biogeographic reconstruction, character evolution, all widely documented in the literature. Here, instead of carefully choosing the sampling and outgroups to have a representative hypothesis, the outgroup is literally what was available in GenBank and the ingroup is highly skewed, given the trans-continental distribution of the genus and the number of species. Without having a wider sampling of the genus and a more careful selection of outgroups, including the sister group of the genus, the calibration has no value.

Please read some of the literature about time-calibration and the choices involved in it. Some examples below:

https://academic.oup.com/sysbio/article/61/2/346/1647963

https://www.ncbi.nlm.nih.gov/pmc/articles/PMC2749537/

https://www.sciencedirect.com/science/article/pii/S1631068313001097

http://phyloworks.org/workshops/DivTime_BEAST2_tutorial_FBD.pdf(this is a very elucidative tutorial).

Author Response

Manuscript ID: plants-876632

Title: Phylogenetic and Comparative Analyses of Complete Chloroplast Genomes of Chinese Viburnum and Sambucus (Adoxaceae)

COMMENTS TO AUTHOR

SPECIFIC RESPONSES TO Reviewers’ comments (All line numbers mentioned below refer to the revised clean version):

Reviewer: 2

Comments to Author

Dear Editor and Authors,

The manuscript presents a careful analysis of more than 20 chloroplast genomes of Viburnum and Sambucus, with full descriptions of the genomic structure. A plastid phylogeny focusing mostly in Viburnum is also presented, as well as a chronogram. The paper is original and has inherent value, but it is not possible to accept it as is, therefore I am recommending Reconsider after major revision.

Response: Thank you for your thoughtful and detailed comments and recommendations. We believe we have been able to incorporate all of these suggestions into the revised version of our manuscript.

Major comments:

#1) Even though the chloroplast descriptions are good, I have objections about the trees presented in the paper, and highly recommend that the results describing them to be rewritten. Several terms related to phylogenetics are misused and do need to be corrected. Also, the time-calibration analysis presents serious conceptual flaws, and I recommend it to be removed from the paper, as it does not contribute with relevant results/discussion. I present some of the arguments below and also some references to be consulted.

Response: Following the referee’s suggestion, we have rewritten the whole part with regard to phylogenetic analyses (i.e., 2.5. Phylogenetic Relationships; Lines 351-436). In addition, we have removed the time-calibration analysis and related sentences in Abstract, Results and Discussion, and Conclusions, also, the Figure 6 and Table 3 from the text (see below responses to #39).

#2) I think the methodology that the authors used to assemble the chloroplasts very unusual, as they used tools (BLAST) that are not traditionally used for small genome assembly. I worry highly repetitive regions might have been thrown away during the de novo assembly. Having a close reference genome, the authors could have easily used an assembler like BWA or Bowtie2, that are optimized to deal with short-read sequences, or even combined methodologies, such as Fast-Plast. Given the very short N50 found for all samples, I’d like to see a comparison of the authors’ methods and a reference assembly, to confirm the results. I also would like to request that the authors include a column in their Supplemental Material Table 1 indicating the sequencing depth in table 1 (how many times each position in the genome was sequenced), as this is a good indication of the assembly quality.

Response: In fact, the methodology that we used to assemble the chloroplasts (i.e., a combination of reference-guided and de novo assembly approaches) quite common, as which has been widely used in many studies (e.g., Zhang et al., 2016; Lu et al., 2017; Li et al., 2017; Ye et al., 2018; Lu et al., 2018; Liu et al., 2020). The specific steps are as follows:

For each species, approximately 5.0 Gb of raw data were generated with pair-end 150 bp read length. First, paired-end sequence reads were trimmed to remove low-quality bases (Q < 20) and adapter sequences using CLC-quality trim tool (quality_trim software included in CLC ASSEMBLY CELL package, http://www.clcbio.com/ products/clc-assemblycell/) before undertaking sequence assembly. Second, the contigs were assembled using CLC de novo assembler with the following optimized parameters: bubble size of 98, minimum contig length of 250, mismatch cost of 2, deletion and insertion costs of 3, length fraction of 0.9, and similarity fraction of 0.8. Third, all the contigs were aligned to the reference chloroplast genome using BLAST (http://blast.ncbi.nlm.nih.gov/), and then the aligned contigs (≥ 90% similarity and query coverage) were ordered according to the reference chloroplast genome. Then, contigs longer than 10 kb were aligned with the reference genome to construct the draft chloroplast genome of each species in Geneious software (http://www.geneious.com). Finally, clean reads were remapped to the draft genome sequences and yielded the complete chloroplast genome sequences.

Through de novo assembly, the maximum number of assembled contigs ranged from 61,001 (V. odoratissimum) to 388,130 (V. melanocarpum), with N50 contigs varying from 285 to 399 bp. A total of 165–209 contigs were successfully mapped to the reference plastomes. Average sequencing depth ranged from about 268 × to 517 × (we have included a column indicating the sequencing depth in table S1). Of these 24 samples, the maximum lengths of assembled contigs ranged from 86,517 to 131,778 bp. For each sample, from de novo assembly of sequence reads, we obtained 3–8 contigs (longer than 10 kb) that had a minimum of 29 bp overlapping sequences in each direction and covered the whole plastome. In spite of the short N50 found in all samples, only long contigs (> 10 kb) were selected to construct the draft whole chloroplast genomes. Furthermore, the four junctions between IRs and SSC/LSC in each species were verified by PCR-based sequencing and the results showed that the assembly sequences were totally identical with the PCR amplified fragments. All of the above evidences could demonstrate the high quality of our assembly. Hence, we believe that our assembly results are robust.

We have rephrased the first paragraph of “2.1. Chloroplast Genome Assembly and Features” (Lines 101-121; changes are underlined):

With the Illumina HiSeq 2500 system, we sequenced the plastomes of 21 species of Viburnum and 3 species of Sambucus. Of these samples, through de novo assembly, the maximum number of assembled contigs ranged from 61,001 (V. odoratissimum) to 388,130 (V. melanocarpum), with N50 contigs varying from 285 to 399 bp. Average sequencing depth ranged from about 268 × (S. adnata) to 517 × (V. melanocarpum) (Table S1). Subsequently, through reference-based assembly, a total of 165–209 contigs were successfully mapped to the reference plastomes. Of which, three to eight long contigs (> 10 kb) that were found to be significantly homologous to the reference genome were combined to generate each chloroplast genome, with no gaps found. The four junctions between IRs and SSC/LSC in each species were initially determined on the basis of these long contigs, and then verified by PCR-based sequencing. The results showed that the assembly sequences were totally identical with the PCR amplified fragments, demonstrating the high quality of our assembly. Finally, we obtained 24 whole chloroplast genome sequences without gaps after de novo and reference-guided assembly, and submitted to GenBank with accession numbers MT507585–MT507605 for Viburnum and MT457821–MT457823 for Sambucus (Table S1).

#3) I think the conclusion could be better worked. The claim that this is a major advance in Adoxaceae phylogeny is fickle, as the paper presents a moderately-sampled phylogeny of only one large genus in the family. Also, the authors state that the plastomes were compared with other members of Adoxaceae, but this is not shown in the manuscript at any point.

Response: We have rewritten the Conclusion (Lines 520-537). The comparison of 24 newly obtained plastomes with other members of Adoxaceae was briefly addressed (Lines 152-155, and Lines 195-196).

#4) The manuscript would also benefit of a review of the English language and style, as I found several parts of the text to be confusing or misleading. I’ve pointed out several of them below, but there are more that I haven’t listed.

Response: Done. We have invited a colleague from University of Wisconsin-Madison (USA) for the reviewing of the English language and style, and believe that this manuscript has been much improved.

Please see the point-by-point comments below, the numbers indicate line numbers.

#5) Line 31: clarify what you mean by large and small genera. Just adding a parenthesis saying “small genera (less than 20 species)”, for example, would be enough.

Response: Done (Line 34).

#6) Line 33: it isn’t possible to say this is a clade without having a phylogeny showing it is a monophyletic lineage. Please either include a reference for a phylogeny of the genus, or just use “group” or “genus”.

Response: Done. We have changed the word “clade” to “genus” (Line 36).

#7) Lines 36, 60, etc: I think the use of Latin America to delimit geographic distribution is very weird, given this is a delimitation based on language/culture and not geography, besides the fact that the current definition of Latin America excludes a few countries in Central and South America and the Caribbean that had French/English/Dutch colonization or that are parts of the French territory. Using Central and South America would be more appropriate, please correct throughout the manuscript.

Response: Done. We have changed “Latin America” to “Central and South America” throughout the manuscript (see Lines 13, 39, 69).

#8) Line 37: change “occurred” to “occurring”

Response: Done. We have changed the word “occurred” to “occurring” (Line 41).

#9) Line 51: change “studied species” to “species sampled in phylogenetic studies” or similar, to make the sentence clearer

Response: Done. We have changed the words “studied species” to “species sampled in phylogenetic studies” (Lines 59-60).

#10) Line 62: double space between taxa and richness

Response: Done (Line 70).

#11) Lines 61 to 65: please rewrite the whole sentence that starts with “By contrast….”. It’s very convoluted and hard to follow, and I couldn’t understand if the relationships of Viburnum were not well studied in China or South/Central America and Southeast Asia.

Response: Done. We have rephrased this sentence as follows (Lines 70-72; changes are underlined):

By contrast, China is considered to be one of the hotspots of Viburnum plant taxa richness, a total of 8 sections and c. 73 species have been found in this region [2], nevertheless, phylogenetic relationships of Viburnum species in where have received much less attention.

#12) Line 68: rewrite objective 1. Suggestion: “characterize and compare the cp genomes of Viburnum species belonging to all the eight sessions occurring in China and related taxa in order to gain insights into their evolutionary patterns;”

Response: Done. We have changed the sentence “characterize and compare the cp genomes of all 8 section-level Viburnum species and related taxa in order to gain insights into their evolutionary patterns;”to “characterize and compare the cp genomes of Viburnum species belonging to all the eight sections occurring in China and related taxa in order to gain insights into their evolutionary patterns;” (Lines 75-77).

#13) Line 80: is this number correct? “with N50 contigs varying from 285 to 399 bp”. These contigs look very short for a chloroplast assembly. Subsequently the authors state “The four junctions between IRs and SSC/LSC in each 82 species were initially determined on the basis of these contigs”. If the contigs were that small, how did you identify junctions based on them?

Response: Yes, the number herein is correct. Please see our above responses to #2.

#14) Line 83: the language in this sentence is misleading. What do the authors mean with “the relevant results”? Does it mean there were results that disagreed with what is being shown in the paper?

Response: We have rephrased the first paragraph of “2.1. Chloroplast Genome Assembly and Features” (see above response to #2).

#15) Table 1: the footnote says “numbers in brackets” while the table only contain parentheses. Include authority names for the species listed in the table.

Response: We have rephrased the footnote under Table 1 (Line 107).

#16) Table 2:

Header: correct table header to “Gene group” and “Gene name”

Footnote: correct “Indicates the gene that present only in Sambucus. Indicates the gene that present only in Viburnum.” to something like: “Indicates the gene is present only in Sambucus. Indicates the gene is present only in Viburnum.”

Footnote: correct “pseudogene was represented by Ψ.” to “pseudogene is represented by Ψ.”

Response: Done.

#17) Line 94: remove the “that” and correct: “including a pair of IR regions (26,272–26,564 bp) separating the LSC region (86,430–87,892 bp) and the SSC region (17,674–18,978 bp).”

Response: Done. We have removed the word “that” and changed the word “separated” to “separating” (Lines 124-125).

#18) Line 100: not possible to understand this sentence: “Notably, five genes (i.e., trnM-CAU, trnT-GGU, trnP-GGG, orf188, and lhbA) and three genes (i.e., psbZ, ndhH, and rpl22) were, respectively, presented in Sambucus and Viburnum.” Do the authors mean genes that occur exclusively in one species or other?

Response: We have rephrased this sentence as follows (Lines 140-142; changes are underlined):

Notably, five genes (i.e., trnM-CAU, trnT-GGU, trnP-GGG, orf188, and lhbA) and three genes (i.e., psbZ, ndhH, and rpl22) were only present in Sambucus and Viburnum, respectively.

#19) Line 104: simplify this sentence: “In particular, the rps12 was trans-spliced gene, with the first exon located in the LSC region, and the second and third in the IR regions.”

Response: Done. We have rephrased this sentence as follows (Lines 144-146):

In particular, the rps12 was a trans-spliced gene, with the first exon located in the LSC region, and the second and third in the IR regions.

#20) Line 106: correct: “We also found that the ycf1 gene at the SSC and IRa border was present as a pseudogene in 16 Viburnum species (Table 2), due to incomplete gene duplication, as shown by previous reports [20, 21].”

Response: Done. We have rephrased this sentence as follows (Lines 146-148; changes are underlined):

We also found that the ycf1 gene at the SSC and IRa junction was present as a pseudogene in 16 Viburnum species (Table 2), due to incomplete gene duplication, as shown by previous reports [20, 21].

#21) Line 112: move the “herein” to right after “obtained”.

Response: Done (Line 152).

#22) Line 115: there’s absolutely no basis for this affirmation: “This is expected given the similarities of morphological characters among them.” There is no direct correlation between the similarities in chloroplast structure to morphology. One good example is family Asteraceae, the family has more than 25 thousand species and the plastomes usually have less than 1% divergence between them. Please remove.

Response: Done. We have removed the sentence “This is expected given the similarities of morphological characters among them.” from the manuscript.

#23) Line 122: correct: “the IRb/SSC and IRa/LSC junction regions,”

Response: We have rephrased this sentence as follows (Lines 180-181; changes are underlined):

The trnN-GUU/ndhF and rpl2/trnH-GUG genes were detected around the IRb/SSC and IRa/LSC junction regions, respectively.

#24) Line 122: decide between using junction or border. Junction is more appropriate.

Response: Following the referee’s suggestion, we have changed the word “border” to “junction” throughout the text (Lines 146, 158, 179, 181-182, 184, 186 ).

#25) Line 124: This sentence is misleading: “Although the boundaries of these species were highly conserved”. Change species to “genomic regions” or similar.

Response: We have rephrased this sentence as follows (Lines 182-184; changes are underlined):

Although the boundaries of these genomic regions were highly conserved, we still observed minor differences between the two genera.

#26) Lines 125, 126: simplify this sentence.

Response: We have rephrased this sentence as follows (Lines 184-186; changes are underlined):

At the LSC/IRb junction, except for V. rhytidophyllum, the IRb regions expanded by 32 bp and 116 bp toward the rps19 gene of the remaining Viburnum species and Sambucus species, respectively.

#27) Line 127: split this sentence in two: “The ndhF gene crossed over the IRb/SSC junction in Viburnum cinnamomifolium and overlapped with the IRb region by 135 bp. It was located at the SSC region in all other Viburnum and Sambucus species, and the whole length varied from 2187 bp to 2250 bp.”

Response: We have rephrased this sentence as follows (Lines 186-188):

The ndhF gene crossed over the IRb/SSC junction in V. cinnamomifolium and overlapped with the IRb region by 135 bp. It was located at the SSC region in all other Viburnum and Sambucus species, and the whole length varied from 2187 bp to 2250 bp.

#28) Line 130: correct: “Notably, the ndhF gene was found to be inverted in all Adoxaceae [26], possibly due to an early stage of the IR expansion followed by a contraction of the boundary.”

Response: We have rephrased this sentence as follows (Lines 188-190):

Notably, the ndhF gene was found to be inverted in all Adoxaceae [26], possibly due to an early stage of the IR expansion followed by a contraction of the boundary.

#29) Line 133: This doesn’t make sense: “The rpl2 gene exhibited invariable sequences among intragenus species in both Viburnum (1490 bp) and Sambucus (1498 bp).” All species are intragenus (?), besides, it’s hard to understand if the whole gene was totally invariable or just parts of it. Correct to something like: “The rpl2 gene was invariable within species in both Viburnum (1490 bp) and Sambucus (1498 bp).”

Response: We have rephrased this sentence as follows (Lines 192-193):

The rpl2 gene was invariable within species in both Viburnum (1490 bp) and Sambucus (1498 bp).

#30) Line 135: correct the sentence: “In addition, all the trnH-GUG genes within the Adoxaceae species studied here had an equal length of 75 bp except for Viburnum oliganthum (78 bp).”

Response: We have rephrased this sentence as follows (Lines 193-195):

In addition, all the trnH-GUG genes within the Adoxaceae species studied here had an equal length of 75 bp except for V. oliganthum (78 bp).

#31) Line 136: Phenomenon is a weird word here, and the sentence is convoluted. Change for something like this: “Similar IR/SC boundary structures shared among genera has also been reported in previous plastome studies [12, 26].”

Response: We have rephrased this sentence as follows (Lines 195-196):

Similar IR/SC boundary structures shared among Adoxaceae species has also been reported in previous plastome studies [12, 26].

#32) Line 145: change “like” to “similarly to most…”

Response: Done. We have changed the word “like” to “similarly to” (Line 200).

#33) Lines 143-165: this whole paragraph could be summarized as a table. Give the Pi values for the genes in both genera in a table and just cite this table in the text. You can reduce this paragraph and give only the highest and lowest values for each genus.

Response: We have summarized the Pi values in a new table (i.e., Table 3) for the top ten most variable coding and noncoding regions in Viburnum and Sambucus, and also, simplified this paragraph (Lines 198-271).

#34) Lines 166-175: there are several things that need to be changed in this paragraph. First, it’s quite confusing trying to understand what the authors mean by “interspecies level” and “species set”. In the first sentence, I’d say to change it to “different taxonomic levels”. In the second case, just mention the markers were used in several studies with Viburnum. The paragraph also contain long sentences, that could be edited.

The most serious thing, though, is the confusion in the meaning of deep and shallow levels of the phylogeny. Traditionally these words are used to distinguish between the backbone and the tips of the tree, with “deep” being used with the meaning of “deeper in time”. Therefore, deep = closer to the backbone, shallow = closer to the tips. This confusion in the meaning is also found in other parts of the manuscript, such as line 23 in the abstract, and 55-58 in the Introduction.

I suggest the authors consult different papers about phylogeny to clarify and understand better this issue, such as https://link.springer.com/article/10.1186/s12862-016-0769-y,

https://onlinelibrary.wiley.com/doi/abs/10.2307/1223970,

https://onlinelibrary.wiley.com/doi/abs/10.1111/j.1365-3113.2006.00355.x?casa_token=yiMTxaqNU0MAAAAA:aCKAp-NggwaPkbLYkFEnx-aOUqro_HgFZm7HhJJhVunfqnQRFL3EK3Cvk1q7AzQ0Nr3ZZLH2tOcr4HM.

Please correct.

Response: Done. Following the referee’s constructive suggestions, we have removed the confusing words “deep” or “deeper” throughout the text and rephrased this paragraph as follows (Lines 272-281; changes are underlined):

Chloroplast DNA region has already been used in exploring the phylogenetic structure and phylogeographic patterns at different taxonomic levels. For instance, hypervariable regions of cpDNA (e.g., matK, ndhF, rbcL, petB-petD, rpl32-trnL, trnC-ycf6, trnH-psbA, trnK intron, and trnS-trnG) were used to infer phylogenetic relationships for several studies with Viburnum [17–19]. Despite increased levels of confidence were revealed in most of the early branches, the relationships within clades of closely related species were still poorly resolved. Most regions used in these studies are today considered low to intermediately variable regions with low Pi values (Figure 3). And additionally, only rpl32-trnL is among the most informative regions of the plastome for most groups (Table 3). Thus, additional phylogenetically informative markers should be included to enhance phylogenetic resolution in low-level phylogenetic or phylogeographic studies.

#35) Line 179: Don’t use “Sambucus plants”, use “Sambucus species”. Correct in the whole manuscript.

Response: Done. We have changed “plants” to “species” throughout the text (Lines 186, 188, 286).

#36) Line 211: correct “supports” to “support”

Response: Done (Line 379).

#37) Lines 212-214: The two clades in the small clade are reciprocal sister-clades, please correct the sentence.

Response: We have rephrased this sentence as follows (Lines 379-382):

The small clade included the genera Sambucus, Adoxa, Tetradoxa, and Sinadoxa, within which samples of Sambucus formed a monophyletic clade (bootstrap percentage, BP = 100%) and were sister to the Adoxa-Tetradoxa-Sinadoxa group.

#38) Lines 210-238: I have several issues with this paragraph. Given the way the phylogeny is described, I have the impression the authors are not familiar with phylogenetic analysis and the jargon involved. One example is the use of “basal”. This has been a long debate in systematics, given that the term is misinterpreted most of the times (see examples: https://pubmed.ncbi.nlm.nih.gov/16701355/, https://onlinelibrary.wiley.com/doi/full/10.1111/j.0307-6970.2004.00262.x). I don’t understand the need to name two subclades, when you could refer directly to the section names.

One serious issue here is the incorrect use of taxonomic nomenclature. Sections should ALWAYS be written with the correspondent generic name, they are part of the binomial. E.g.: Viburnum sect. Tomentosa,Viburnum sect. Tinus, etc. Please refer to the INC for details (https://www.iapt-taxon.org/nomen/main.php). This needs to be corrected in the figure 5 too. When using informal names, such as clades Sambucina, etc., usually these names are not italicized, as they don’t have a formal rank.

This whole paragraph needs to be rewritten, taking into account the explained above, and also to simplify the text. As it is right now, it’s a wall of text, without any breaks. Please re-read and reformulate, mainly using shorter sentences and maybe splitting the paragraph in two.

Response: Following the referee’s suggestion, we have rewritten the whole part with regard to phylogenetic analyses (i.e., 2.5. Phylogenetic Relationships; Lines 370-455). In addition, sections were written with a binomial method when they firstly appeared in the text (Lines 24-25, 27, 396, 400-401), such as Viburnum sect. Viburnum. While after that, to be clearer, they were written with shorter forms, such as section Viburnum. Similar usage has been found in previous studies (e.g., Winkworth and Donoghue, 2004, 2005; Clement and Donoghue, 2011; Clement et al., 2014)

#39) Figure 6 and Lines 246-267: I don’t understand the need to include a time-calibrated tree in this study, especially with the conceptual problems present in the analysis. The dating conducted in this study is highly questionable given the choice of outgroups and calibration point, and I strongly recommend completely removing this part from the paper.

The choice of outgroup and ingroup sampling in a phylogeny can highly impact and bias results of comparative analysis such as calibration, biogeographic reconstruction, character evolution, all widely documented in the literature. Here, instead of carefully choosing the sampling and outgroups to have a representative hypothesis, the outgroup is literally what was available in GenBank and the ingroup is highly skewed, given the trans-continental distribution of the genus and the number of species. Without having a wider sampling of the genus and a more careful selection of outgroups, including the sister group of the genus, the calibration has no value.

Please read some of the literature about time-calibration and the choices involved in it. Some examples below:

https://academic.oup.com/sysbio/article/61/2/346/1647963

https://www.ncbi.nlm.nih.gov/pmc/articles/PMC2749537/

https://www.sciencedirect.com/science/article/pii/S1631068313001097

http://phyloworks.org/workshops/DivTime_BEAST2_tutorial_FBD.pdf(this is a very elucidative tutorial).

Response: Indeed, the lack of rigorous protocols for assigning calibrations based on

fossils and selecting outgroups and ingroups would raise serious questions about the credibility of divergence dating results according to the above-mentioned literatures, hence, following the referee’s very helpful suggestions, we decided to remove the time-calibration analysis and all associated results from the paper (see above responses to #1).

Reviewer 3 Report

Ran et al. have contributed plastome data from 24 endemic species in Adoxaceae (21 Viburnum and 3 Sambucus), and they have delicately investigated genomic structure, gene order and content, SSRs etc potential genetic resources from these data. Finally they combined other 20 plastome data from GenBank proving a 46-taxa phylogeny and time tree for Chinese Adoxaceae species, aiming to fill the gap in understanding of relationships in the woody flowering plant clade Viburnum.

It is a generally well written and formatted paper. They have analyzed quite amount of data, and a serials of analyses are well done. But I do have several concerns:

  • . The abstract can be further improved. The first two sentences are trying to set a tone that they want to reconstruct a phylogeny of Viburnum globally, because previous studies are “only in a regional context”. Well, after I read through, the present study only sampled the Chinese species, isn’t also regional? Somehow the results/conclusion part in the abstract is too general to be meaningful. It would be better to highlight your results and findings.

  • . Why the authors did not combine cp dataset of the 22 species from Clement et al. (2014; ref 19) to make a global phylogenetic analyses of Vibunum? I think the phylogenetic placement and  endemsm of Chinese lineages are more objective if you put the sampling in a global scale. Plus the data is published/available. If it’s not the goal to do so, make sure clarify, because it’s confusing as current layout.

  • Please make sure the terms used in this study consist and accurate across the whole text. For examples:
  1. in the “Results and Discussion” section, you mentioned “de novol assembly”, whereas in the M&M section you said “a combination of de novo and reference-guided”. Please clarify.
  2. P9L224: “the earliest diverging lineage” I think this expression is not correct: in a bifurcation tree, two sisters diverged on the MRCA node at the same time, why one is the earliest?

  • The authors explored throughout the plastome data(e.g., 2.3. Sequence Divergence Analysis) and advocated some phylogenetically informative markers, why not apply in this study to make an excellent example? Instead, the four data matrices used to build the phylogeny described in M&M, I don’t think these matrices are significantly different and represented.

  • Please justify why you sampled a hybird in your phylogeny? Why Viburnum erosumis not monophyletic in the phylogeny (see Figures 5, 6)?

  • I wish the authors input more discussions on those subclades and sections recognized in the phylogeny. If they agree with FOC, what morphological characters they share? I think these kind of discussion will make the paper more interesting and botanically informative.

  • These Viburnum species are endemic to China, is it possible for author to provide images (flower, fruit, etc) for some representatives from certain Viburnum and Sambucusgroups and aligning on the side of the phylogeny tree?

  • For Figure 6, there are a few comments:
  1. The figure seems not complete; timescale needs to be provided at x -axis with geoscale labels; (K/T) boundary line need to be marked in the tree
  2. Better detailed figure legend needed; all the elements in the plot should be explained. For instance, what those letters and numbers in the black circles mean? Are they corresponding the subclades and sections in Figure 5?
  3. I expect the authors to label the Figure 6 with those sections as Figure 5 (e.g.,subclades, sections, etc ), and discuss when those clades diveged in the time tree?

Please find the rest detailed comments in the attached pdf file.

Miao Sun

Author Response

Manuscript ID: plants-876632

Title: Phylogenetic and Comparative Analyses of Complete Chloroplast Genomes of Chinese Viburnum and Sambucus (Adoxaceae)

COMMENTS TO AUTHOR

SPECIFIC RESPONSES TO Reviewers’ comments (All line numbers mentioned below refer to the revised clean version):

Reviewer: 3

Comments to Author:

Ran et al. have contributed plastome data from 24 endemic species in Adoxaceae (21 Viburnum and 3 Sambucus), and they have delicately investigated genomic structure, gene order and content, SSRs etc potential genetic resources from these data. Finally they combined other 20 plastome data from GenBank proving a 46-taxa phylogeny and time tree for Chinese Adoxaceae species, aiming to fill the gap in understanding of relationships in the woody flowering plant clade Viburnum.

It is a generally well written and formatted paper. They have analyzed quite amount of data, and a serials of analyses are well done. But I do have several concerns:

#1) The abstract can be further improved. The first two sentences are trying to set a tone that they want to reconstruct a phylogeny of Viburnum globally, because previous studies are “only in a regional context”. Well, after I read through, the present study only sampled the Chinese species, isn’t also regional? Somehow the results/conclusion part in the abstract is too general to be meaningful. It would be better to highlight your results and findings.

Response: Done. We have rephrased the Abstract (Lines 10-30).

#2) Why the authors did not combine cp dataset of the 22 species from Clement et al. (2014; ref 19) to make a global phylogenetic analysis of Viburnum? I think the phylogenetic placement and endemism of Chinese lineages are more objective if you put the sampling in a global scale. Plus the data is published/available. If it’s not the goal to do so, make sure clarify, because it’s confusing as current layout.

Response: In fact, we tried to combine the chloroplast genomes of the 22 species from Clement et al. (2014), but could not download the complete annotated nucleotide data set in TreeBASE (S15758) and Dryad (doi:10.5061/dryad.hh12b) for some reasons. Also, the 73 regions (including 22 coding regions and 51 noncoding regions) published in GenBank for the 22 species could not align together with our dataset easily, as many of which do not have GenBank accession numbers as they are either too short (> 200 bp) to be accepted by GenBank, duplicate previously published data, or have been annotated as part of coding regions. More importantly, just as the reviewer said, our goal was to reconstruct a phylogeny of Viburnum in a regional text (i.e., in China) rather than a global phylogenetic analysis of Viburnum, hence, we decided not to combine cp dataset of the 22 species. In addition, following the referee’s very helpful suggestions, we have rephrased the Abstract part (Lines 10-30) and some related sentences throughout the text to make the layout of the paper more reasonable (Lines 70-72, 75-78, 540-541, 554-556).

#3) Please make sure the terms used in this study consist and accurate across the whole text. For examples:

in the “Results and Discussion” section, you mentioned “de novol assembly”, whereas in the M&M section you said “a combination of de novo and reference-guided”. Please clarify.

Response: We have revised the first paragraph of “2.1. Chloroplast Genome Assembly and Features” (Lines 101-121; changes are underlined), trying to make the assemble methodology of these 24 chloroplast genomes clearer.

With the Illumina HiSeq 2500 system, we sequenced the plastomes of 21 species of Viburnum and 3 species of Sambucus. Of these samples, through de novo assembly, the maximum number of assembled contigs ranged from 61,001 (V. odoratissimum) to 388,130 (V. melanocarpum), with N50 contigs varying from 285 to 399 bp. Average sequencing depth ranged from about 268 × (S. adnata) to 517 × (V. melanocarpum) (Table S1). Subsequently, through reference-based assembly, a total of 165–209 contigs were successfully mapped to the reference plastomes. Of which, three to eight long contigs (> 10 kb) that were found to be significantly homologous to the reference genome were combined to generate each chloroplast genome, with no gaps found. The four junctions between IRs and SSC/LSC in each species were initially determined on the basis of these long contigs, and then verified by PCR-based sequencing. The results showed that the assembly sequences were totally identical with the PCR amplified fragments, demonstrating the high quality of our assembly. Finally, we obtained 24 whole chloroplast genome sequences without gaps after de novo and reference-guided assembly, and submitted to GenBank with accession numbers MT507585–MT507605 for Viburnum and MT457821–MT457823 for Sambucus (Table S1).

P9L224: “the earliest diverging lineage” I think this expression is not correct: in a bifurcation tree, two sisters diverged on the MRCA node at the same time, why one is the earliest?

Response: Following the referee’s suggestion (Reviewer 2), we have rewritten the whole part with regard to phylogenetic analyses (i.e., 2.5. Phylogenetic Relationships; Lines 370-455).

#4) The authors explored throughout the plastome data (e.g., 2.3. Sequence Divergence Analysis) and advocated some phylogenetically informative markers, why not apply in this study to make an excellent example? Instead, the four data matrices used to build the phylogeny described in M&M, I don’t think these matrices are significantly different and represented.

Response: In the present analysis, the phylogenetic tree was constructed only based on the complete chloroplast genome sequence dataset. In fact, we tried to reconstruct the phylogenetic tree using several of those (i.e., 6 regions) highly variable regions listed in Table 3, but the topology was a little different, and the bootstrap values of some main nodes were not high. Maybe it is not easy to select suitable regions or combinations of regions for the phylogenetic analysis. One of the main goals of this study was to screen and identify repeat sequences, simple sequence repeats (SSRs) and mutational hotspot regions for future species identification and phylogeographic studies of the two genera. Hence, we identified the mutational hotspot regions that would be potentially useful markers for future phylogenetic and phylogeographic analysis, but did not apply them in this study. Perhaps our future studies will focus on assessing the ability of the proposed 20 highly variable regions (Table 3) to discriminate relationships among sections, subsections and closely related species. To make the layout of the paper more reasonable, we have removed the PIC analysis and results from the text, and rephrased some related sentences throughout the text (Lines 17-20, 78-80, 271, 499).

#5) Please justify why you sampled a hybird in your phylogeny? Why Viburnum erosum is not monophyletic in the phylogeny (see Figures 5, 6)?

Response: The hybrid Viburnum carlesii × Viburnum macrocephalum resulted from the hybridization between Viburnum carlesii and Viburnum macrocephalum, and which was grouped to the section Viburnum. Since both of its parents were recognized as members of section Viburnum in Flora of China, thus, we suppose that the sampling of this hybrid would have little effect on the phylogeny of Viburnum. But for all this, we decided to remove the hybrid from the new phylogenetic analysis to avoid any kind of controversy.

In this study, we found that phylogenetic studies have uniformly and strongly supported earlier recognized sections and subsections, while proven difficult to resolve recent divergences within groups of closely related species based on both limited parsimony informative sites (Clement et al., 2014; ref19) and whole chloroplast genome sequences (this study). Though samples of Viburnum fordiae, Viburnum dilatatum and Viburnum erosum formed a monophyletic clade (BP = 100%), the relationships among them were poorly resolved (BPs < 90%), hence, based on the phylogenetic tree herein (Figure 5), we could hardly know the real relationships among the three species, and also, we could not confirm the monophyly of Viburnum erosum species. Maybe in the future, a number of nuclear loci or the phylogeny with more individuals of Viburnum erosum sampled could give us the answer.

#6) I wish the authors input more discussions on those subclades and sections recognized in the phylogeny. If they agree with FOC, what morphological characters they share? I think these kinds of discussion will make the paper more interesting and botanically informative.

Response: Following the referee’s suggestion (Reviewer 2), we have rewritten the whole part with regard to phylogenetic analyses (i.e., 2.5. Phylogenetic Relationships; Lines 370-455).

#7) These Viburnum species are endemic to China, is it possible for author to provide images (flower, fruit, etc) for some representatives from certain Viburnum and Sambucus groups and aligning on the side of the phylogeny tree?

Response: Sorry, we would like to do such a thing, but it is a pity that few high-quality images are available in our camera.

#8) For Figure 6, there are a few comments:

  • The figure seems not complete; timescale needs to be provided at x -axis with geoscale labels; (K/T) boundary line need to be marked in the tree
  • Better detailed figure legend needed; all the elements in the plot should be explained. For instance, what those letters and numbers in the black circles mean? Are they corresponding the subclades and sections in Figure 5?
  • I expect the authors to label the Figure 6 with those sections as Figure 5 (e.g., subclades, sections, etc), and discuss when those clades diveged in the time tree?

Response: The lack of rigorous protocols for assigning calibrations based on fossils and selecting outgroups and ingroups could raise serious questions about the credibility of divergence dating results. Without having a wider sampling of the genus Viburnum and a more careful selection of outgroups, including the sister groups of the genus, it is likely that the calibration has no value in this study. Hence, following the referee’s very helpful suggestions (i.e., Reviewer 2), we have removed the time-calibration analysis and related sentences in Abstract, Results and Discussion, and Conclusions, also, the Figure 6 and Table 3 from the paper.

Minor suggestions:

#9) Lines 12-13: “these phylogenetic relationships were evaluated only in a regional context” If this study is only focus on the a group of Viburnum endemic to China, then it’s also fit the description “evaluated only in a regional context”, unless you sampled all the representive lineages globally.

Response: In this study, our goal was to reconstruct a phylogeny of Viburnum in a regional text (i.e., in China) rather than a global phylogenetic analysis of Viburnum, hence, following the referee’s very helpful suggestions, we have rephrased the Abstract part and some related sentences throughout the text to make the layout of the paper more reasonable (see above responses to #2).

#10) Lines 21-24: “Phylogenomic analyses based on different chloroplast genome datasets suggested almost consistent relationships within Adoxaceae and Viburnum with a previous combined analysis of 113 species, and provided important insights into the deep phylogenetic relationships and diversification events of the eight sections of Viburnum recognized in Flora of China.” I would like to see the authors highlight what is the significant founding in this study? rather that repeat what others already reported? Or just provide the cp genome data to the community?

Response: Done. We have rephrased the Abstract (Lines 10-30).

#11) Lines 66-67: “we reported whole-plastome sequence data for 21 species of Viburnum, covering all of the 8 currently diagnosed sections in Flora of China,” Why not combined cp dataset of the 22 species from the ref 19, make a global phylogentic analyese of Vibunum? I think the phylogentic placement and its endemsm are more objective if you put the sampling in a global scale. Plus the data is available. If the authors did so, at least mentioned here briefly; Otherwise it is not well informed, since the M&M section is located in the end.

Response: In this study, our goal was to reconstruct a phylogeny of Viburnum in a regional text (i.e., in China) rather than a global phylogenetic analysis of Viburnum, hence, we decided not to combine cp dataset of the 22 species. In addition, following the referee’s very helpful suggestions, we have rephrased the Abstract part and some related sentences throughout the text to make the layout of the paper more reasonable (see above response to #2).

#12) Line 79: “de novo assembly,” In method, it said a combination of de novo and reference-guided, please confirm.

Response: We have revised the first paragraph of “2.1. Chloroplast Genome Assembly and Features, trying to make the assemble methodology of these 24 chloroplast genomes clearer (see above response to # 3).

#13) Lines 83-84: “The relevant results showed that the assembly sequences were totally identical with the PCR amplified fragments, demonstrating the high quality of our assembly.” How taxonomic ids of each of the samples are confirmed? How to rule out they are the same species?

Response: Four junction regions between IRs and SSC/LSC in each plastome were first determined based on de novo contigs, and subsequently confirmed by PCR amplifications and Sanger sequencing using specific primers (Table S2). The specific steps are as follows:

Specific primers for the four junction regions were developed for Viburnum and Sambucus, respectively. Then, PCR amplifications and Sanger sequencing were conducted on each sample individually, with PCR products for each region of each sample numbered. After that, each numbered sequence was aligned with its corresponding assembled plastome sequence to validate the identity between the amplified fragments targeting the four junctions and the four junction regions determined based on de novo contigs.

#14) Figure 2: I would suggest the authors align the species names to left in Figure 2, if possible.

Response: Done.

#15) Line 135: “trnH-GUG” Please clarify why trnH-GUG region is missing from V. cinnimomifolium in Figure 2? What is the possible reason? Does this have evolutionary implication or meaning?

Response: We are sorry to make you misunderstand, in fact, the trnH-GUG region is not missing. Because the ndhF gene crossed over the IRb/SSC junction in V. cinnamomifolium while located at the SSC region in all other Viburnum and Sambucus species, thus, we thought there was no need to mark the location of the trnH-GUG region from V. cinnimomifolium in Figure 2. Well, to be more understandable, we have added the location of the gene trnH-GUG from V. cinnimomifolium to Figure 2.

#16) Line 144, 163, 174, 185, 187: “Figure A1, Figure A2, Figure A3, Figure A4” Please clarify if this is the supplementary.

Response: Done. We have added a prefix of “See Appendix A” prior to “Figure A1, Figure A2, Figure A3” throughout the text (Lines 199, 346, 353, 366). In addition, we have removed the original Figure A2 from the text.

#17) Lines 163-165, 174-175: “Future chloroplast-based phylogenetic and phylogeographic studies would benefit greatly from the application of these highly variable markers.”

“Thus, additional phylogenetically informative markers should be included to enhance phylogenetic resolution in low-level phylogenetic or phylogeographic studies.”

It feels that this sentence is redundant with the one in Line 164 and Line 165

Response: We have removed the sentence “Future chloroplast-based phylogenetic and phylogeographic studies would benefit greatly from the application of these highly variable markers.” (Line 271) from the text.

#18) Line 209: “Phylogenetic Relationships” For the phylogeny part, are V1 and V2 subclades also recongized by FOC? Are there mophorlogical characters to support these two subclades? How the V1 and V2 disntinct in mophorlogy? Are all the 8 sections sampling corvered in this study monophyletic? Please add more discussion.

Response: Following the referee’s suggestion (Reviewer 2), we have rewritten the whole part with regard to phylogenetic analyses (i.e., 2.5. Phylogenetic Relationships; Lines 370-455.

#19) Line 214: “Adoxa-Tetradoxa-Tetradoxa” what is the meaning of this group? which study does this name follow? why repeated “Tetradoxa” used?

Response: Sorry, these were genuine mistakes, we have changed Tetradoxa in lines 361, 363 to Sinadoxa.

#20) Line 218: “and “*” indicates 100% support values in ML.” If so, why there are still 100 in the outgroups?

Response: Sorry, these were genuine oversights, and we have changed 100% over the branches comprising Tetradoxa, Adoxa and Sinadoxa to “*” in Figure 5.

#21) Line 224: “earliest diverging lineage” I think this expression is not corret: in a bifurcation tree, two sisters divegent at the MRCA node at the same time, why one is the earliest?

Response: Following the referee’s suggestion (Reviewer 2), we have rewritten the whole part with regard to phylogenetic analyses (i.e., 2.5. Phylogenetic Relationships; Lines 370-455; see above responses to #3).

#22) Line 242: “Figure 6” Please explain in the figure caption, what those letters and numbers in the black circles mean. Are they corresponding the subclades and sections in Figure 5? Please clarify. Also need to cross referenced as Table 3. I expect the authors label the Figure 6 with those sections as Figure 5, and discuss when those clades diveged in the time tree? Timescale needs to be provided at x -axis with geoscale labels.

Response: The Figure 6 and Table 3 have been removed from the paper (see above response to #8).

#23) Line 252: “Cretaceous/Tertiary (K/T) boundary” This is usually estimated about 66 Ma. I suggest to label/mark this in the time tree. easier for readers to use as reference, since it has been mentioned in the discussion.

Response: The time-calibration analysis and related sentences throughout the text have been removed from the munuscript (see above responses to #8).

#24) Line 253: “These inferred divergence times were slightly older than those previously reported” The authors said the ages slightly older than those previously reported, is this becuase of minimum age constraint on node A? Have the authors explore the dating senario if node A is constraint as max age? Why not compare with ages estimated from ref 26, since it’s relevant to this study and sharing the same fossil constraint?

Response: The time-calibration analysis and related sentences throughout the text have been removed from the munuscript (see above responses to #8).

#25) Lines 264-267: “it would be likely that the diversification of these major lineages within Viburnum probably have originated coincided with a global cooling and drying period during which much of the Earth’s history have been shaped since the end of the Eocene [42].” How this conclusion is made? because of the time frame? or did the authors conduct the correlation?

Response: The time-calibration analysis and related sentences throughout the text have been removed from the munuscript (see above responses to #8).

#26) Line 315: “MAFFT multiple aligner v7” what options are used in MAFFT?

Response: We used the plugin MAFFT in GENEIOUS v11.1.4 with default settings (Line 536) for fast alignment of multiple chloroplast genome sequences.

#27) Lines 326-327: “20 plastomes downloaded from the GenBank, and two outgroups Panax ginseng and Eleutherococcus nodiflorus (Table S3)” Southeast Asia (6 species)? How are the 20 plastomes from GenBank represented in terms of taxonmy and geography? In TableS3, I have no.11 is a hyrid. Would the authors please provide the justification why you sampled the hybird? What is the placement of this hybird? what support? Is there phylogentic conflict for this hybird among those different plastid markers?

Response: In this study, 19 Adoxaceae plastomes (the hybrid was deleted) were downloaded from the GenBank, including 6 plastomes of section Odontotinus, 1 plastome of section Megalotinus, 3 plastomes of section Solenotinus, 2 plastomes of section Viburnum, plus 7 representatives of Sambucus, Adoxa, Sinadoxa, and Tetradoxa. These selected plastomes represent all major lineages of Chinese Adoxaceae that could be downloaded. Notably, only very limited number of Viburnum plastomes are available in GenBank and we have downloaded all of the plastomes which belonging to the eight sections recognized in Flora of China.

We have removed the hybrid from the new phylogenetic analysis to avoid any kind of controversy (see above responses to #5).

#28) Lines 329-330: “(1) complete cp genomes; (2) complete cp genomes without missing loci; (3) protein-coding sequences; (4) protein-coding sequences without missing loci.” Why use these four data sets? Are there fundimantaly differences in phylogenetic signals among the four data sets? In “2.3. Sequence Divergence Analysis”, the authors repeatly highlight certain regions/loci are suitable for phylogentic study, why not try out in this study inferring the relastionships for Viburnum? I really would like to see that results.

Response: Please see our above responses to #4.

#29) Lines 330-333: “Chloroplast sequences of these 46 plants were aligned together using MAFFT. ML analysis was conducted in RAXML-HPC [52] on the CIPRES cluster (http://www.phylo.org/), with a GTR+G+I substitution model selected by jModelTest v2.1.7 [53]. is the matrix partitioned when running the RAxML?

Response: The ML analysis were performed for the complete chloroplast genome sequence dataset with an unpartitioned strategy. It has been revealed that the ML analyses based on different partitioning strategies would produce highly congruent phylogenetic topologies [e.g., Lu et al., 2017. The Complete Chloroplast Genomes of Three Cardiocrinum (Liliaceae) Species: Comparative Genomic and Phylogenetic Analyses; Li et al., 2017. Comparative Genomics and Phylogenomics of East Asian Tulips (Amana, Liliaceae); Yao et al., 2019. Plastid phylogenomic insights into the evolution of Caryophyllales]. Thus, we believe that the ML analysis would yield nearly identical tree topologies based on both partitioned and unpartitioned matrices in this study. We have specified that the ML analysis was conducted in RAXML-HPC for the complete chloroplast genome sequence dataset with an unpartitioned strategy (Line 538).

Round 2

Reviewer 2 Report

Dear authors,

Thank you for revising the manuscript, I feel it's much improved. I specially like the deeper dive into the phylogeny and comparisons with previous studies. Also, I like that the methods are clarified and the inclusion of a new table. I only have a few grammar corrections to suggest, which are specified below. Also, I noticed that bootstrap support is being abbreviated as BP, when the usual acronym is BS, so I suggest changing that too. 
About the use of section names, I understand the omission of the genus name for the sake of brevity, and that it has been done before, nevertheless, still goes against the rules of the nomenclature code. The section name is part of the the binomial, and has no validity without the genus name. It's a minor thing, and I will refer to the editor to request a change or not. 

Grammar corrections:

252: change “that comprising” to “that comprises”

253: Change “The first clade Crenotinus was marked by curving

(crenate) leaf teeth [19],” to “The first clade Crenotinus, marked by curving

(crenate) leaf teeth [19],”

256: Change “and also the sister relationships between which and section Tomentosa.” to “and also the sister relationship between this section and section Tomentosa.”

257: Change “The second clade was Nectarotinus [19], which marked by extrafloral nectaries, contained” to “The second clade was Nectarotinus [19], which is marked by extrafloral nectaries, containing”

262: Change “monophyletic section of Tinus” to “monophyletic section Tinus”

263: change “it” to the section name, it will be clearer

284: change non-monophyly to non-monophyletic and the “have” to “has”

Author Response

Manuscript ID: plants-876632

Title: Phylogenetic and Comparative Analyses of Complete Chloroplast Genomes of Chinese Viburnum and Sambucus (Adoxaceae)

COMMENTS TO AUTHOR

SPECIFIC RESPONSES TO Reviewers’ comments (All line numbers mentioned below refer to the revised clean version):

Reviewer: 2

Comments to Author

Dear authors,

Thank you for revising the manuscript, I feel it's much improved. I specially like the deeper dive into the phylogeny and comparisons with previous studies. Also, I like that the methods are clarified and the inclusion of a new table. I only have a few grammar corrections to suggest, which are specified below. Also, I noticed that bootstrap support is being abbreviated as BP, when the usual acronym is BS, so I suggest changing that too.

Response: Following the referee’s very helpful suggestions, we have changed “BP” to “BS” throughout the text (Lines 211, 213, 224, 229, 235, 239 and 245).

About the use of section names, I understand the omission of the genus name for the sake of brevity, and that it has been done before, nevertheless, still goes against the rules of the nomenclature code. The section name is part of the the binomial, and has no validity without the genus name. It's a minor thing, and I will refer to the editor to request a change or not.

Response: We have changed the section names using a binomial method throughout the text. Please see the new munuscript.

Grammar corrections:

#1) 252: change “that comprising” to “that comprises”

Response: Done (Lines 225-226).

#2) 253: Change “The first clade Crenotinus was marked by curving (crenate) leaf teeth [19],” to “The first clade Crenotinus, marked by curving (crenate) leaf teeth [19],”

Response: Done (Line 226).

#3) 256: Change “and also the sister relationships between which and section Tomentosa.” to “and also the sister relationship between this section and section Tomentosa.”

Response: Done (Line 230).

#4) 257: Change “The second clade was Nectarotinus [19], which marked by extrafloral nectaries, contained” to “The second clade was Nectarotinus [19], which is marked by extrafloral nectaries, containing”

Response: Done (Line 231).

#5) 262: Change “monophyletic section of Tinus” to “monophyletic section Tinus”

Response: Done (Line 235).

#6) 263: change “it” to the section name, it will be clearer

Response: Done (Line 237).

#7) 284: change non-monophyly to non-monophyletic and the “have” to “has”

Response: Done (Lines 258-259).

Reviewer 3 Report

This is the second time I read this manuscript (ms), and authors have done a great job in improving this ms. I'm satisfied with their responding, except:

(corresponding to the remark index from the 1st round review)

#4. It's not necessary to remove whole analysis at all, since you have already done. I think you still can provide such potential genetic resources to the community, but with caveats, and the relevant text may go to the supplementary.

#5. If the authors keen to resolve these relationships, maybe target enrichment methods with custom designed probes for Viburnum is the way to go.

#8. It's pity to see the authors remove the whole dating section, because of lack of rigorous protocols.

#27. Please make sure these information also reflect in the text to readers.

Other comments:

I really like the updated "2.5. Phylogenetic Relationships" section, would the authors please to consider replacing "was marked by" with "was characterized by" or something similar? I think it's more appropriate for describing morphological characters when using "characterized" or "distinguished" (just some food for thoughts). Plus "was marked by" is used too much.

In Figure 5, the substitution rate unit is missing in the phylogeny. "Outgroup" should also be labeled out.

Last, the authors need to update the Table S3, at least the hybrid still presents in it. It's good opportunity to make sure all the updates are consistently reflected across the whole manuscript. Thanks!

Miao

Author Response

Manuscript ID: plants-876632

Title: Phylogenetic and Comparative Analyses of Complete Chloroplast Genomes of Chinese Viburnum and Sambucus (Adoxaceae)

COMMENTS TO AUTHOR

SPECIFIC RESPONSES TO Reviewers’ comments (All line numbers mentioned below refer to the revised clean version):

Reviewer: 3

Comments to Author:

This is the second time I read this manuscript (ms), and authors have done a great job in improving this ms. I'm satisfied with their responding, except:

(corresponding to the remark index from the 1st round review)

#1) #4. It's not necessary to remove whole analysis at all, since you have already done. I think you still can provide such potential genetic resources to the community, but with caveats, and the relevant text may go to the supplementary.

Response: Sorry, we still decided to remove these analyses from the text, since the results seemed to make no sense in this study.

#2) #5. If the authors keen to resolve these relationships, maybe target enrichment methods with custom designed probes for Viburnum is the way to go.

Response: Thanks for your helpful suggestions, maybe we would like to resolve these relationships using this method in the future.

#3) #8. It's pity to see the authors remove the whole dating section, because of lack of rigorous protocols.

Response: Sorry, a robust molecular dating analysis is important for the discussion and conclusion, to make the manuscript more rigorous, we decided to remove the whole dating section following the referee’s suggestion (i.e., Reviewer 2).

#4) #27. Please make sure these information also reflect in the text to readers.

Response: Yes, please see Lines 323-326.

Other comments:

#5) I really like the updated "2.5. Phylogenetic Relationships" section, would the authors please to consider replacing "was marked by" with "was characterized by" or something similar? I think it's more appropriate for describing morphological characters when using "characterized" or "distinguished" (just some food for thoughts). Plus "was marked by" is used too much.

Response: Done, we have changed the word “marked” to “characterized” in Line 226 and Line 231.

#6) In Figure 5, the substitution rate unit is missing in the phylogeny. "Outgroup" should also be labeled out.

Response: Done. Please see the new Figure 5.

#7) Last, the authors need to update the Table S3, at least the hybrid still presents in it. It's good opportunity to make sure all the updates are consistently reflected across the whole manuscript. Thanks!

Response: Done. Please see the new Table S3.